# Three-dimensional printing of silica glass with sub-micrometer resolution

Po-Han Huang [1,5], Miku Laakso[1,5], Pierre Edinger[1], Oliver Hartwig [2], Georg S. Duesberg[2], Lee-Lun Lai[1], Joachim Mayer[3], Johan Nyman [4], Carlos Errando-Herranz [1], Göran Stemme[1], Kristinn B. Gylfason [1] & Frank Niklaus [1] ✉

Silica glass is a high-performance material used in many applications such as lenses, glassware, and fibers. However, modern additive manufacturing of micro-scale silica glass structures requires sintering of 3D-printed silica-nanoparticle-loaded composites at ~1200 °C, which causes substantial structural shrinkage and limits the choice of substrate materials. Here, 3D printing of solid silica glass with sub-micrometer resolution is demonstrated without the need of a sintering step. This is achieved by locally crosslinking hydrogen silsesquioxane to silica glass using nonlinear absorption of sub-picosecond laser pulses. The as-printed glass is optically transparent but shows a high ratio of 4-membered silicon-oxygen rings and photoluminescence. Optional annealing at 900 °C makes the glass indistinguishable from fused silica. The utility of the approach is demonstrated by 3D printing an optical microtoroid resonator, a luminescence source, and a suspended plate on an optical-fiber tip. This approach enables promising applications in fields such as photonics, medicine, and quantum-optics.

Modern life is unthinkable without glass, with applications ranging from glassware and windows to optical lenses and fibers. Silica glass has excellent material properties such as thermal and chemical stability, hardness, and optical transparency in a wide wavelength range. Yet, as a result of the stability and the brittleness of silica glass, fabricating three-dimensional (3D) silica glass objects with features at the micrometer scale remains challenging, although structures at this scale are critical for many exciting applications, for example, in nanophotonics[1], nanoelectromechanical systems, and nanofluidics[2]. To address this, additive manufacturing of silica glass by stereolithography[3,4], direct ink writing[5–7], digital light processing[8], and multiphoton polymerization[9–11] have been explored. Moreover, hybrid approaches that combine multiple manufacturing techniques and silica sources have been recently investigated[12]. Although 3D structures made of high-quality silica glass have been demonstrated, these

approaches can, at best, resolve feature sizes of several tens of micrometers[13], except for a recent study that has reported sub-micrometer resolution[11]. Moreover, the application of these approaches is limited by the mandatory high-temperature processing since they are all based on sol-gel methods using different organic mixtures loaded with up to 50 wt.% of silica nanoparticles to achieve desired rheological or photochemical properties. As a result, the as-printed materials are composites with a large content of organics and discrete silica nanoparticles that do not feature silica glass properties. After 3D printing, sintering of the printed materials at temperatures of around 1200 °C is necessary to obtain functional solid and transparent silica glass. The mandatory sintering process at such elevated temperatures severely limits the application space and integration compatibility of these methods. This is because any substrate materials or pre-manufactured structures onto which the 3D-printed silica-glass

[1]Division of Micro and Nanosystems, School of Electrical Engineering and Computer Science, KTH Royal Institute of Technology, Stockholm 10044, Sweden. [2]Institute of Physics, Faculty of Electrical Engineering and Information Technology, University of the Bundeswehr Munich & SENS Research Center, Neubiberg 85577, Germany. [3]Central Facility for Electron Microscopy (GFE), RWTH Aachen University, Aachen 52074, Germany. [4]Department of Physics, Chemistry and Biology (IFM), Linköping University, Linköping 58183, Sweden. [5]These authors contributed equally: Po-Han Huang, Miku Laakso. ✉e-mail: frank@kth.se

structures are to be directly integrated must withstand the thermal treatment, which essentially eliminates most materials of interest. In other cases, final assembling of the 3D-printed structures and other substrates or structures required for the application would be necessary, which can be exceedingly challenging for structures at the scale of micrometers.

Hydrogen silsesquioxane (HSQ) is an inorganic silica-like material described by the empirical formula HSiO$_{1.5}$[14]. HSQ has been widely utilized as a high-resolution negative-tone resist that can be patterned by electron beams[15], ion beams[16], and deep ultraviolet (UV) light with a wavelength below 248 nm[17]. These conventional patterning techniques are all based on cross-linking HSQ by linear-absorption of electrons, ions, or photons, for realizing two-dimensional (2D) patterns. Several methods for fabricating suspended structures in HSQ using linear-absorption methods have been demonstrated by varying the depth of cross-linking of the HSQ by locally tuning the energy of the electron or ion beam[16,18]. However, it is not feasible to fabricate free-form 3D structures using these methods. Recently, cross-linking of HSQ using sub-picosecond laser by nonlinear absorption of photons has been investigated[19], while the demonstrated structures were still limited to suspended 2D beams, and the appearance of silica-glass chemical bonds in the crosslinked material was not demonstrated.

In this work, we report a process for 3D printing of silica glass that is solid and optically transparent as-printed and features sub-micrometer resolution. In this process, we take advantage of our finding that hydrogen silsesquioxane without any additives can be selectively crosslinked into silica glass in 3D by exposure to sub-picosecond laser pulses with a wavelength of 1040 nm, which is a nonlinear absorption process as HSQ has no linear sensitivity to light with wavelengths above 248 nm[17]. We show by Raman, energy-dispersive X-ray (EDS), and photoluminescence spectroscopy that the as-printed material is silica glass but, compared to fused silica glass, features a higher ratio of 4-membered silicon-oxygen rings in the network resulting from sub-picosecond laser exposure, photo-luminescence, residual hydrogenated and hydroxyl species, and trace amounts of organic residuals. These features and residuals can be removed by a 900 °C annealing step, resulting in a low shrinkage of 6.1% of the 3D-printed structures and an increase of the hardness and reduced elastic modulus of the 3D-printed silica glass to values expected for fused silica glass. We demonstrate that the as-printed silica glass is of good quality and suitable for application in micro-optics and that the differences in the optical performance of the 3D-printed silica glass are insignificant before and after annealing. Our results will inspire many applications in important fields of science and technology, including cell biology, chemistry, quantum optics, and photonics, by making silica glass with its superior properties available on the size scale of sub-micrometer to micrometers with the capability of integrating the glass structures onto a variety of substrates.

## Results

### 3D printing of silica glass by nonlinear cross-linking of HSQ

Our process for 3D printing of silica glass consists of three main steps (Fig. 1a–c): (1) drop casting of HSQ dissolved in organic solvents onto a substrate, (2) tracing the desired 3D shape in the dried HSQ with the focus of the sub-picosecond laser beam, and (3) dissolving the unexposed HSQ using a potassium hydroxide solution. Using this process, we produced transparent silica glass structures with high patterning fidelity, smooth sidewalls, and sub-micrometer features (Fig. 1d–i). The smallest voxel dimensions we obtained were ~65 nm in width and 260 nm in height, featuring an aspect ratio of 4 (Supplementary Fig. 1). The printed material is silica glass, i.e., amorphous silicon dioxide, as confirmed by electron diffraction (Fig. 1k) and EDS. EDS data collected from the bulk of the as-printed material showed an elemental composition consisting of silicon and oxygen along with a residual atomic concentration of carbon of below one percent (Supplementary

Table 1). The printed silica glass is without porosity at least down to the scale of a few nanometers, which was the lowest observable feature size when inspecting cross-sections of printed structures using scanning transmission electron microscopy and scanning electron microscopy (SEM) (Fig. 1j and Supplementary Fig. 2). In contrast to stereolithography and direct ink writing, our 3D printing process does not rely on organic compounds, acting as photoinitiators or binders, which then remain in the printed material. Instead, our process relies on direct cross-linking of inorganic HSQ, having the empirical formula of HSiO$_{1.5}$[14], to silica glass by sufficient local nonlinear absorption of photons of a sub-picosecond laser with a wavelength of 1040 nm.

### Material characterization of the 3D-printed silica glass

To identify the chemical bonds in the 3D-printed silica glass, we collected its Raman spectrum, which showed all the features expected for silica glass (Fig. 2a). We observed three additional features in the spectrum of the as-printed silica glass, which originate from residual hydrogen, residual carbon, and 3- and 4-membered rings in the silica glass network. These features could be removed by annealing the as-printed silica glass at 900 °C in air, thus making its Raman spectrum indistinguishable from that of a commercial fused silica glass substrate (Fig. 2a). To elucidate the effects of annealing, we collected Raman spectra from printed silica glass annealed at 150 °C, 300 °C, 500 °C, and 800 °C (Fig. 2a and Supplementary Fig. 3a). The identified hydrogen-related species included Si-H bonds, hydroxyl groups (OH), and molecular water. The presence of Si-H bonds indicates incomplete cross-linking of HSQ. The Si-H Raman signal disappeared already after annealing at 150 °C, which is consistent with results reported for annealing thin spin-coated HSQ films[20]. Hydroxyl groups and molecular water are often found in silica glasses with high water content[21]. Their Raman signals, together with those of the carbon species, disappeared after annealing at 900 °C, which correlates well with earlier reports of silica glass precursors substantially decreasing their water and hydroxyl signals after annealing at 800 °C[22], and carbon-containing silicon oxides starting to lose their carbon content already at 500 °C[23,24]. The only chemicals containing carbon in the entire printing process were the two solvents in which the HSQ was dissolved (methyl isobutyl ketone and toluene), and the surfactant (Triton) that is typically used in the development process to minimize the effects of bubbles. Since HSQ itself does not contain carbon[14] and we did not use the surfactant in the samples prepared for the material characterization to eliminate it as a possible carbon source, we hypothesize that the residual carbon species originated from the solvents that might not have entirely evaporated from the 3D-printed silica glass. Exposure by femtosecond laser pulses has been reported to change the ratio between the 3- and 4-membered rings in silica glass[25,26], which is in line with our observation of the differences in the corresponding Raman features between the spectrum of our as-printed glass and that of commercial silica glass. In addition, exposure of silica glass to femtosecond laser has been reported to cause atomic-scale structural changes that give rise to photoluminescence in the glass[25–27]. This agrees well with our observation that a sloped photo-luminescence background is present in the Raman spectra of our glass samples that were annealed at temperatures between 150 °C and 800 °C (Fig. 2a and Supplementary Fig. 3a). We characterized the background by collecting a spectrum covering the entire background peak of a sample annealed at 500 °C. This spectrum revealed that the background observed in the Raman spectra is part of a broad photo-luminescence peak centered at slightly above 2 eV with a long tail towards higher photon energies (Supplementary Fig. 3b). Photo-luminescence at these energies can originate from at least three different types of defects in the silica glass network caused by laser exposure[25–27]. These defects are non-bridging oxygen hole centers with and without hydrogen bonding, respectively causing photo-luminescence peaks at 2.0 eV and 1.9 eV, silicon clusters at 2.2 eV, and

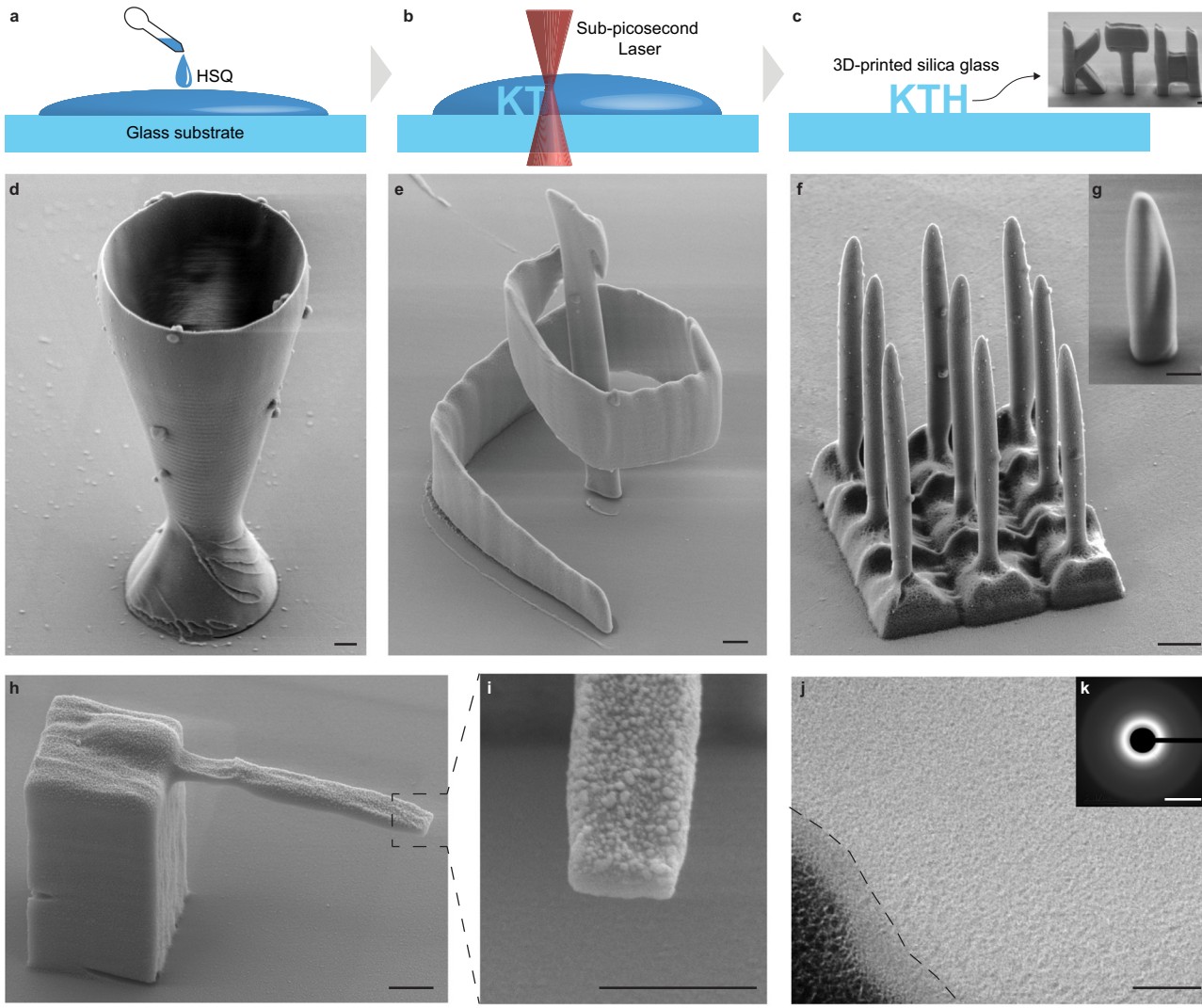

**Fig. 1 | 3D printing of silica-glass micro-structures by direct laser writing.**
**a** Preparation of hydrogen silsesquioxane (HSQ) by drop casting. **b** Direct writing in the HSQ using the focus of the sub-picosecond laser beam. **c** Development of the 3D-printed structures by dissolving the unexposed HSQ. The inset SEM image shows a 3D-printed KTH logo (scale bar, 1 μm). **d, e** SEM images of 3D-printed structures: a glass goblet and a conical spiral (scale bars, 1 μm). **f, g** SEM images of an array of 500 nm diameter needles and of a pillar with a longitudinal twist (scale bars, 1 μm). **h, i** SEM images of a cantilever that is 750 nm wide and 540 nm thick taken from a 45-degree angle. The roughness on the top surface originates from a deposited gold layer for reducing charging effects during imaging (scale bars, 1 μm). **j** Scanning TEM image of an as-printed silica glass structure. The dark region on the bottom-left corner is the metal layer deposited during sample preparation, and the interface between the glass and metal is marked with a dashed line (scale bar, 50 nm). **k** Electron diffraction pattern of the TEM sample. The pattern consists of concentric rings, showing that the printed material is amorphous (scale bar, 5 nm⁻¹).

oxygen-deficiency centers (i.e., direct silicon-silicon bonds) at 2.7 eV[26,28]. The photoluminescence in our samples was removed by annealing at 900 °C, after which the ratio between the Raman peaks of the 3- and 4-membered rings also became comparable to the ratio seen in the spectrum of commercial fused silica glass. This is consistent with an earlier report which showed that the changes in the properties of silica glass, including its photoluminescence, Raman features, and refractive index caused by laser exposure, could be reversed by annealed the laser-exposed silica glass at 900 °C[27].

**Effects of annealing on the 3D-printed structures**
The optical transparency of printed silica glass is important for its use in optical and photonic micro-structures. To show that our 3D-printed glass features a level of optical transparency that is relevant for micro-optical components, we 3D-printed structures that consist of a 3 μm thick suspended glass plate that is placed above a ring on the substrate. We imaged the ring through the plate using optical microscopy, both before and after annealing (Fig. 2c, d). In addition, as reference,

we 3D-printed the same ring structure without the suspended plate (Supplementary Fig. 4). By comparing Fig. 2c and Supplementary Fig. 4, we observed that the ring can be clearly seen through the suspended plate, as is the case for the ring without a suspended plate. The view of the ring through the suspended plate was slightly distorted compared to the view of the ring without a suspended plate. This distortion originates from the surface topography of the suspended plate. The glass-transition temperature of silica glass is 1200 °C[29], and as expected, annealing of our 3D-printed micro-structures at 1200 °C smoothened their features. This improved the topography of the surfaces of the printed suspended plates, thereby minimizing the distortions caused by the plates in the optical microscope images taken through them (Fig. 2e). Smoothening can be further improved by repeating the annealing at 1200 °C (Fig. 2f).

Annealing 3D-printed micro-structures can cause them to shrink, distort, or even collapse[24]. The ultimate linear shrinkage of our 3D-printed silica glass when annealed at 900 °C is only (6.1 ± 0.8)% (Fig. 2b and Supplementary Fig. 5), which compares favorably to shrinkages of

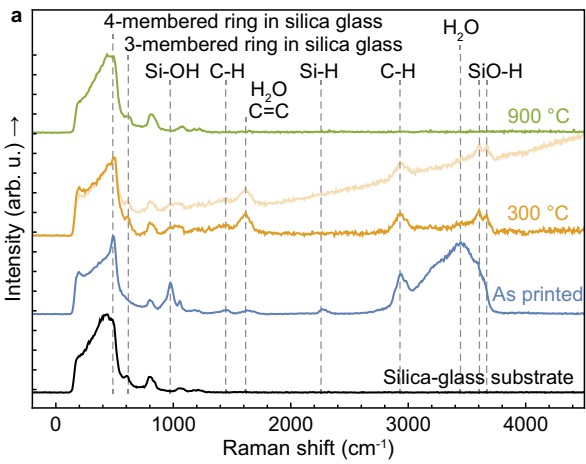

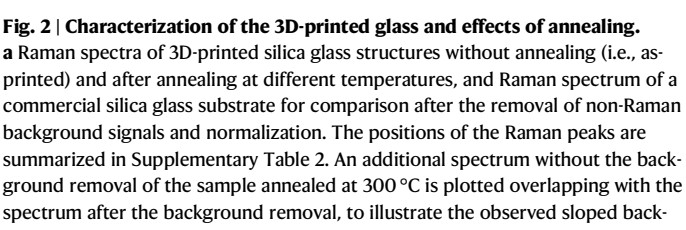

**Fig. 2 | Characterization of the 3D-printed glass and effects of annealing.**
**a** Raman spectra of 3D-printed silica glass structures without annealing (i.e., as-printed) and after annealing at different temperatures, and Raman spectrum of a commercial silica glass substrate for comparison after the removal of non-Raman background signals and normalization. The positions of the Raman peaks are summarized in Supplementary Table 2. An additional spectrum without the background removal of the sample annealed at 300 °C is plotted overlapping with the spectrum after the background removal, to illustrate the observed sloped background. **b** Relative linear shrinkage of the 3D-printed silica glass test structures,

shown in Supplementary Fig. 5b, after annealing at different temperatures. The error bars show the largest measured variation from the mean value or the measurement uncertainty, whichever was larger. Five measurements were performed for each data point. **c** SEM image of a 3D-printed silica glass plate suspended on four legs with a ring structure underneath. Below the SEM image is an optical microscope top view of the ring structure imaged through the suspended plate. **d–f** The same structure as in **c** after further annealing at 900 °C for one hour, 1200 °C for 1 hour, and 1200 °C for 2 hours, respectively. (Scale bars, 1 μm).

between 16% and 56% for glass objects made by stereolithography and direct ink writing methods that require sintering at around 1200 °C in order to remove all the organic content and densify the printed objects[13]. In contrast to these methods, the low shrinkage of our 3D-printed silica glass enables the preservation of the shape of the 3D-printed structures and avoids delamination of the structures from the substrate. To demonstrate this important feature, we printed a suspended plate with four attachment points that were spaced by 10 μm to the substrate and annealed the structure at 900 °C for one hour (Fig. 2d). We observed no cracks in the structure or delamination of the attachment points from the substrate after annealing with the associated material shrinkage of the structure. Further annealing of the 3D-printed glass at 1200 °C caused reflow and surface smoothening of the glass structures. As discussed above, the reflow of the glass does not represent linear material shrinkage. Even in this extreme case, we did not observe cracks or delamination in our 3D-printed glass structures after annealing at 1200 °C for up to 2 hours (Fig. 2b, e, f). To elucidate the origin of the residual 6.1% shrinkage of our 3D-printed structures, considering that our as-printed glass is neither porous nor contains significant carbon, we analyzed the TEM diffraction patterns of the glass before and after annealing, and found that the shrinkage originated from the rearrangement of its amorphous structure and minor amount of crystallization (Supplementary Fig. 6).

The mechanical properties of the 3D-printed silica glass are important for its use in many applications, including nanoelectromechanical systems. With this in mind, we characterized the hardness and reduced elastic modulus of the 3D-printed silica glass by performing nanoindentation measurements on 3D-printed microplates with a footprint of 20 μm by 20 μm and a thickness of ~2 μm. The measured hardness and reduced elastic modulus of the as-printed silica glass were $2.4 \pm 0.2$ GPa and $40 \pm 2$ GPa, respectively. After annealing at 900 °C for 1 h, the hardness and reduced elastic modulus increased to $7.7 \pm 0.6$ GPa and $75 \pm 2$ GPa, respectively, which are almost identical to the values we measured of reference samples consisting of fused-silica microplates of the same geometry (Supplementary Table 3 and Supplementary Fig. 7). The differences in the hardness and reduced elastic modulus between the as-printed glass and the fused silica reference sample are

expected since we have observed residual hydrogen, hydroxyl groups, and water content in the Raman spectrum of the as-printed glass, and since the as-printed glass shrinks by up to $(6.1 \pm 0.8)\%$ upon annealing (Fig. 2a, b). Our observation that the hardness and reduced elastic modulus of the 3D-printed glass after annealing at 900 °C reached the values of the fused-silica microplate reference samples, suggests that the residual components in our 3D-printed glass were completely removed, and that the glass was fully densified at an annealing temperature of 900 °C. This is also consistent with our observations that the Raman spectrum of the 3D-printed glass after annealing is indistinguishable from that of fused silica, and that the material shrinkage reached its ultimate value of 6.1% after annealing at 900 °C.

## 3D-printed optical microtoroid resonator in silica glass
Our approach for 3D printing of transparent silica glass can be used to realize functional photonic microsystems, which we demonstrated by printing an optical microtoroid resonator coupled to an integrated photonic bus waveguide (Fig. 3a, b). The 3D design freedom of the printing process allowed us to print the bus waveguide with couplers slanted upwards from the substrate plane, which enabled convenient out-of-plane coupling of light between the ends of the waveguide and external optical fibers. Furthermore, the design freedom also allowed us to suspend the entire system at least 3 μm above the substrate surface, thus preventing optical coupling of the light into the substrate. We characterized the 3D-printed resonator by measuring its transmission spectra in the optical telecommunication S, C, and L bands between 1460 nm and 1580 nm when injecting the fundamental transverse magnetic ($TM_{00}$) and electric ($TE_{00}$) mode into the bus waveguide. To monitor any possible effects of annealing on the functionality of the resonator, we characterized the resonator as printed, as well as after annealing at 150 °C, 300 °C, and 900 °C, respectively. The spectra measured before annealing and after annealing at 900 °C with the $TM_{00}$ mode injection are shown in Fig. 3c, and the spectra for other annealing temperatures and $TE_{00}$ mode injection are shown in Supplementary Fig. 8a, b. All measured transmission spectra, with and without annealing, showed clear resonances, thus confirming that the resonator works as expected. The free spectral range (FSR, i.e.,

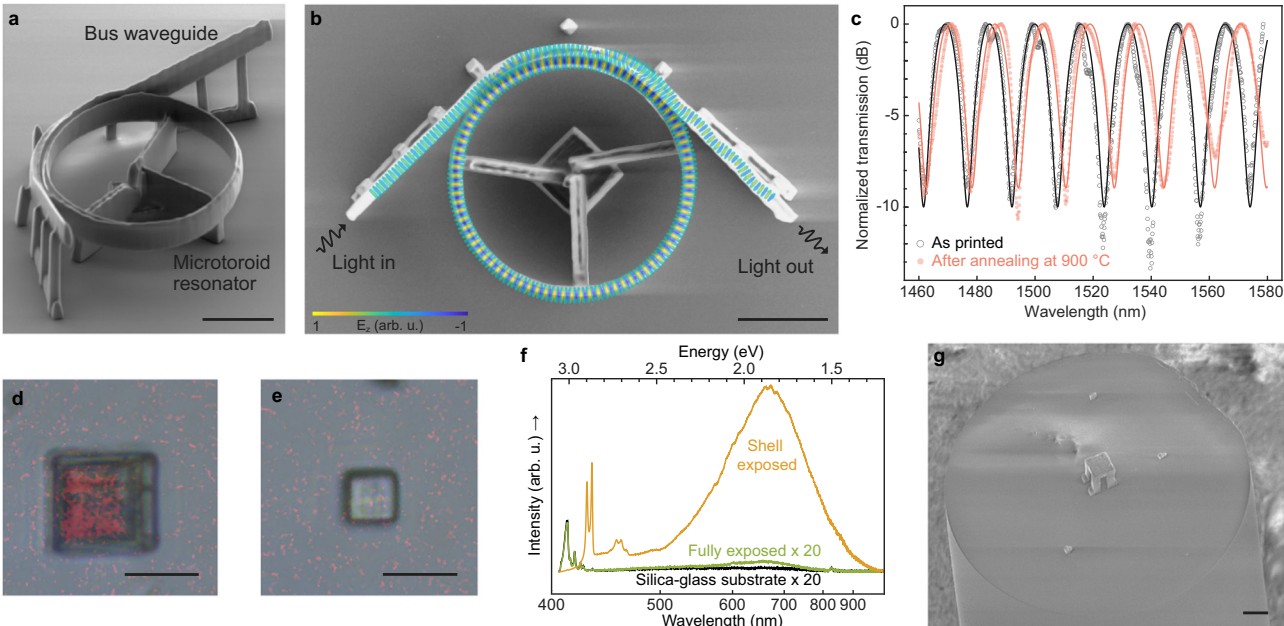

**Fig. 3 | 3D-printed optical demonstrators and their characterization. a** SEM image of a 3D-printed photonic microtoroid resonator with a bus waveguide without annealing (i.e., as-printed). **b** SEM top view of the same device as in **a** superimposed with the simulated vertical component of the electrical field ($E_z$) at resonance when injecting $TM_{00}$ mode into the bus waveguide (Supplementary Fig. 9). Original images used for the superimposition are shown in Supplementary Fig. 10. **c** Measured transmission spectra of the resonator in **a** without annealing and after annealing at 900 °C when injecting dominantly $TM_{00}$ mode, normalized to their respective maximum values. Solid lines show the results of a resonator model fitted to the measured transmission spectra. **d**, **e** Optical microscope image with top view of two cubes exposed by laser pulses only at its shell and through the entire volume, respectively, after annealing at 1200 °C, superimposed with the corresponding photoluminescence images. Original images used for the superimposition are shown in Supplementary Fig. 13. **f** Photoluminescence spectra of the core of the shell-exposed cube in **d** the fully exposed cube in **e** and the silica glass substrate. The intensity of the displayed spectra of the fully exposed cube and the substrate is scaled up by a factor of 20 for improved clarity. **g** SEM image of a suspended glass plate 3D-printed onto the tip of an optical fiber. (Scale bars, 10 μm).

separation between resonances) was between 16 nm and 17 nm without annealing and trended slightly upwards as the annealing temperature was increased (Supplementary Fig. 8c). These results match well with the expected FSR value of 16 nm before annealing and its increase to 17 nm after annealing at 900 °C, predicted by Eq. (1) in the Methods using simulated group indexes (Supplementary Fig. 9), and assuming a linear shrinkage of the circumference of the resonator of 6.1% due to annealing (Fig. 2b). We measured a maximum loaded quality factor of 500 for a single resonance peak (at 1540 nm in Fig. 3c), from which we estimated a resonator round-trip loss between 1.8 and 2.5 dB (Supplementary Fig. 11). This round-trip loss can include scattering losses due to sidewall-roughness, geometrical discontinuities such as the directional coupler and waveguide anchors, waveguide bend loss, and material loss (Supplementary Fig. 12). Among these, the losses related to the printing technology itself, that is, the sidewall-roughness scattering loss and the material loss, are expected to be low. This is because the measured sidewall-roughness of our 3D-printed glass structures is as low as 1 nm (Supplementary Fig. 12c), and the annealing did not affect the quality factor of the resonator (Supplementary Fig. 8d), respectively. The apparent variations in the quality factor of peaks in one spectrum and across spectra measured after annealing the resonator at different temperatures were due to the possibility of the resonator to support multiple modes (Supplementary Fig. 9). The evaluation of the quality factor of a peak is sensitive to the shape of the peak which depends on the distribution of optical power in different modes. The optical power distribution is in turn sensitive to changes in the coupling condition between the optical fibers and the waveguide. Thus, the resonator was functional and its FSR was stable and predictable across all evaluated annealing temperatures without significant effects of annealing on its quality factor, demonstrating that the 3D-printed silica glass can be used for photonic

and optical micro-devices, both with and without subsequent annealing step.

## Selective formation of Si nanocrystals in printed structures

Photoluminescence sources, in contrast to the laser-induced defects discussed above, can also be intentionally embedded in HSQ, by generating silicon nanocrystals using high-temperature annealing of non-laser-exposed HSQ[30]. Thus, by combining laser patterning and annealing, our 3D printing process enables selective functionalization of the 3D-printed structures for luminescence applications. We demonstrated this by printing two cubes on a substrate, one of which was a laser-exposed shell encapsulating a core of unexposed HSQ, while the other had its entire volume exposed to laser (Fig. 3d, e and Supplementary Fig. 13). After annealing of the cubes at 1200 °C in air, a strong photoluminescence peak, centered at a wavelength of 670 nm (1.85 eV), and several Raman features at below 4000 cm$^{-1}$ that do not belong to silica glass were observed in the volume of the unexposed HSQ (Fig. 3f and Supplementary Fig. 14). The photoluminescence peak indicates the presence of silicon nanocrystals, while the laser-exposed shell, as well as the fully laser-exposed cube, showed little to no photoluminescence (Fig. 3d–f). In addition to the full freedom of embedding silicon nanocrystals inside printed silica glass structures in 3D, the properties of silicon nanocrystals are also tunable by manipulating annealing parameters and environments[30]. This protocol paves a unique way towards applications that utilize silicon nanocrystals, including light-emitting devices, nonlinear optics, photovoltaic cells, and sensors[31,32].

## 3D printing of silica glass on optical-fiber tips

Finally, to demonstrate the integration flexibility of our approach enabled by direct printing of optically transparent and functional

silicas glass without a need for sintering, we printed a suspended silica glass plate on the tip of an optical fiber with temperature-sensitive protective polymer coatings and jackets (Fig. 3g). We found that the 3D-printed glass plate was perfectly aligned to the light-guiding core of the fiber as designed, and the temperature sensitive protective polymer coatings and jackets of the fiber were preserved after printing.

## Discussion

Taken together, the results in the present work show that our 3D printing technology makes it possible to additively manufacture transparent and solid 3D silica glass structures with sub-micrometer features on a substrate surface without the need of thermal post-processing. These capabilities are going well beyond the capabilities of existing surface micromachining techniques, including those that utilize growth, deposition, lithography, etching, and liftoff of silica glass layers and those that use direct cross-linking of HSQ via linear-absorption of electrons or deep UV light[16–18]. These techniques are capable of manufacturing only 2D or suspended planar structures. Although cross-linking of HSQ by nonlinear absorption of photons has been investigated[19], no free-form 3D structures have been demonstrated. On the other hand, existing 3D manufacturing methods of silica glass are severely limited in terms of design flexibility, integration, and applicable substrate materials. While laser-defined wet etching is a subtractive manufacturing process defining the 3D structures inside a bulk substrate[33–35], sol-gel-based methods allow additive 3D manufacturing but require a mandatory sintering step at high temperatures of around 1200 °C to form a glass from the 3D-printed composite material and to obtain viable optical properties[3–10]. In contrast, our approach allows integrating 3D silica glass structures with excellent optical functionality without thermal post-processing onto substrates that contain pre-manufactured micro-structures and that cannot withstand elevated temperatures, as demonstrated by 3D printing a suspended silica glass plate on the tip of an optical fiber with polymer coatings. The capabilities of our approach could be further extended by coating the 3D-printed micro-structures with metals or other functional materials, thus tailoring the properties of the final 3D structure[36,37], or by mixing functional materials into HSQ before printing[38]. For example, introducing nanodiamonds[39] would enable hybrid quantum photonics integration[40–42] and adding ferrous nanoparticles[43] could achieve magnetically remote motion control of the printed structures. With these enabling capabilities and the wide range of promising extensions, our glass 3D printing technology will find applications in fields such as photonics[1,44], quantum optics, fluidics[2], 3D-printed microelectromechanical systems[45], robotics, cell biology[46], and chemistry. While beyond the scope of this work, further investigation of the mechanism behind multiphoton cross-linking of HSQ to silica glass using sub-picosecond laser pulses will be of interest for both research and applications. Such investigations could provide insights that may help to reduce the high ratio of 4-membered silicon-oxygen rings and avoid photoluminescence in the as-printed glass, and that could contribute to a deeper understanding of the light-matter interaction in transparent materials.

## Methods

### Drop-casting HSQ on a substrate
Two types of glass substrates were used in the experiments. For most experiments, a silica glass substrate (JGS2 optical-grade fused quartz, MicroChemicals) with a thickness of 250 μm was chosen. These substrates were optically transmissive in the wavelength range between 270 nm and 2 μm and have a typical hydroxyl concentration of below 300 parts per million. For the experiments in which a high-magnification oil-immersion objective was used, the substrate was a borosilicate cover slip (Thermo Scientific) with a thickness of 170 μm. In all experiments, the substrates were first cleaned by rinsing with acetone and then with isopropanol, followed by drying under airflow.

Next, HSQ solution (FOX16, Dow Corning, USA) which contains methyl isobutyl ketone and toluene as the solvents was drop-casted on the substrate. The HSQ layer was grown to a thickness of about 100 μm by drop-casting multiple times on the same location while allowing for a few minutes of drying in a fume hood at room temperature between the casts. After drop-casting, the samples were left to dry at room temperature in a fume hood overnight (~8–12 hours). After drying the solvents, a hard HSQ layer was left on the substrate.

### 3D printing by direct laser writing
The dried HSQ on the glass substrate was exposed using a sub-picosecond laser (Spirit 1040-4-SHG, Spectra-Physics of Newport Corporation) operating at a central wavelength of 1040 nm, a repetition rate of 10 kHz, and a pulse duration of 298 fs. The laser was focused through the glass substrate in the HSQ. The substrate, with the dried HSQ on top of it, was moved by a 3-axis linear motorized stage (XMS100, Newport) and the movement speed during printing was typically between 0.5 μm/s and 1 μm/s. Two alternative objectives were used for focusing: (1) an objective with a numerical aperture of 0.65 (Plan Achromat RMS40X, Olympus) operated without immersion oil or (2) an objective with a numerical aperture of 1.4 (Objective Plan-Apochromat 63x/1.4 Oil DIC, Carl Zeiss) operated with immersion oil (Immersol, Carl Zeiss). The first objective was used for the larger structures while the second objective with the higher numerical aperture was used for manufacturing some of the structures with sub-micrometer resolution. Suitable laser powers for exposing the HSQ were found by observing the appearance of the patterned structures in the HSQ through the objective using a camera. The single-pulse energies used in the patterning were between 14 nJ and 18 nJ when using the first objective and around 7.5 nJ when using the second objective. The energies were measured with a silicon optical power detector (918D-SL-OD3R, Newport) after the pulses exited the objective.

### Development of the 3D-printed structures
The HSQ on the glass substrate that was not exposed to the laser light was removed in a development step. The development was done by immersing the sample in a 0.1 M solution of potassium hydroxide (Sigma-Aldrich) in de-ionized water. To this mixture, 0.05 vol% of Triton X-100 (LabChem Inc.) was added as a surfactant to decrease the size of bubbles formed in the development process, thus reducing the impact and potential damage that the bubbles can cause to the 3D-printed micro-structures. The development was done for at least 8 hours and thereafter the sample was rinsed with de-ionized water. Most of the samples were left to dry in air at room temperature, but the sample shown in Fig. 1f was dried using critical-point drying to prevent breaking of the structures by surface tension.

### Annealing of 3D-printed samples
For the 3D-printed glass samples that were annealed, annealing was done in an oven (Metallwarenfabrik 51/s, Conrad Naber, Germany) in an air atmosphere. Samples were placed inside the oven when it was at room temperature. Afterwards, the oven was heated to the target temperature with maximum heating power of the oven and kept at the annealing temperature for one hour, after which the oven was powered off and left to cool down naturally, resulting in a defined temperature ramping profile (Supplementary Fig. 15). The samples were removed from the oven only after it had cooled to below 150 °C.

### Preparation and inspection of sample cross-sections
Cross-sections of 3D-printed glass samples were prepared by cutting through the as-printed structures using focused ion beam (FIB) milling (Strata FIB 400 dual beam, FEI Company). Two kinds of samples were prepared: A cross-sectional cut for inspection by SEM (Supplementary Fig. 2), and thin lamellas for inspection using field emission TEM (JEM-

F200, JEOL Ltd.) (Fig. 1j and Supplementary Fig. 6). The outermost layers of the lamellas were removed by low-energy ion polishing to avoid possible artefacts[47]. Energy-dispersive X-ray spectroscopy was conducted on lamella to study their elemental compositions with a silicon drift detector (X-Max, Oxford Instruments). The EDS spectrum was measured at 11 points of the lamella, and a summary of the results is presented in Supplementary Table 1.

## Raman spectroscopy and photoluminescence

Raman spectroscopy and photoluminescence experiments were conducted using a confocal Raman microscope (alpha 300R, WITec) equipped with 405 nm and 532 nm wavelength lasers coupled to the microscope using a single-mode optical fiber. The 405 nm laser was used for all measurements except for the comparison spectrum in Supplementary Fig. 14 which was measured with the 532 nm laser. The laser power was manually set to below 5 mW, which prevented thermal damage to the samples, while simultaneously maintaining a reasonable measurement time. The collected light was guided to a fiber-coupled 300 mm ultrahigh-throughput spectrometer (UHTS 300, WITec). A 600 g/mm grating was used to disperse the collected light onto a CCD camera. This setup provided an energy resolution of below 3 cm$^{-1}$, which is suitable for Raman measurements as well as for the collection of broad photoluminescence signals. The microscope was equipped with an objective that had a magnification of 100x and a numerical aperture of 0.9. This objective in the confocal microscope suppressed the signals from the focal planes about 0.5 μm above and below the focus point. The 3D-printed glass structures from which Raman and photoluminescence spectra were collected each consisted of a block, a few micrometers in size, suspended between 7 μm and 9 μm above a substrate surface by supporting pillars (Supplementary Fig. 3c). This spacing, together with the vertical resolution of the microscope, allowed gathering of Raman and photoluminescence spectra from the 3D-printed glass blocks without interfering contributions from the substrates.

## Photoluminescence-superimposed image

Each image was constructed of a bright-field image and a photoluminescence image both taken at the same position using the bright-field and fluorescence modes of an optical microscope (Axio Scope.A1, Carl Zeiss), respectively. Under the fluorescence mode, a light-emitting diode module with a wavelength of 470 nm (Carl Zeiss) and a filter set with an excitation wavelength of 470/40 nm and an emission wavelength of 525/50 nm (Filter set 38 Endow GFP shift free, Carl Zeiss) were used. The photoluminescence image was processed and then superimposed onto its corresponding bright-field image using ImageJ.

## Shrinkage of 3D-printed samples due to annealing

Five 3D-printed T-shaped structures that were placed at a defined spacing on the substrate (Supplementary Fig. 5b) were used for characterizing shrinkage of the silica glass as a result of annealing. The lengths of the horizontal beams of the T-shaped structures were measured using SEM before annealing and after annealing at each temperature. The fixed spacing between the vertical pillars of the T-shaped structures was used for calibrating all SEM measurements to compensate for focusing and sample orientation differences between the measurements. The mean values of the relative length decrease of the five beams from their original lengths gave the relative linear shrinkages shown in Fig. 2b. The error bars shown in Fig. 2b are either the largest measured variation from the mean at each temperature or the propagated error originating from scaling and measurement accuracy uncertainties, whichever was larger. The relative length changes of the individual T-shaped test structures are shown in Supplementary Fig. 5a.

## Nanoindentation characterization

Three types of samples were characterized. The first sample type consists of a microplate with a footprint of 20 μm × 20 μm and a thickness of ~2 μm, 3D-printed on a fused-silica substrate. The second sample type consists only of a flat fused-silica substrate for reference. Finally, to consider the potential effects of the geometry of the plates on the indentation measurement results, we fabricated a third sample type by engraving microplates of the same dimensions as the 3D-printed plates into a fused-silica substrate using laser ablation. In all experiments, the fused-silica substrates were JGS2 optical-grade fused quartz purchased from MicroChemicals GmbH. For the indentation measurements, we used two specimens of each of the three sample types, one of the specimens was annealed at 900 °C for one hour, and the other specimen was not annealed. A nanoindenter (Hysitron TI 950 Triboindenter, Bruker) was used to measure the hardness and reduced elastic modulus of the samples via the Oliver–Pharr method[48]. The indenter was equipped with a Berkovich tip and was calibrated using a standard quartz reference sample with a hardness and elastic modulus of 9.3 and 69.6 GPa, respectively. All measurements were run in the load-controlled mode with set maximum loads that yielded indents that are sufficiently deep to avoid tip radius influence and do not exceed 200 nm, that is, not >10% of the thickness of the microplates, to avoid contributions from the substrate. On each sample, three measurements were performed.

## Microtoroid resonator design

The silica glass microtoroid resonator was designed using both 2D eigenmode waveguide simulations using Lumerical MODE 2020a software (Supplementary Fig. 9), and 3D finite difference time domain device simulations using Lumerical FDTD 2020a software (Fig. 3b). The waveguide cross-section was ellipsoidal due to the voxel shape of the writing laser system, and based on the simulations, the cross-section height was designed to be 2.5 μm and the width 1.2 μm. These dimensions are sufficiently large to confine the light and thus minimize scattering caused by the supports of the bus waveguide and the microtoroid resonator. According to our electromagnetic simulations, the waveguide dimensions simultaneously limit the number of supported wave modes to three dominantly transverse electric (quasi-TE) and three dominantly transverse magnetic (quasi-TM) modes in the optical communications between 1460 nm and 1580 nm (Supplementary Fig. 9). The free spectral range FSR of a microtoroid resonator can be computed using the equation

$$FSR = \frac{\lambda^2}{n_g 2\pi r}, \qquad (1)$$

where $\lambda$ is the wavelength of the light, $n_g$ is the group index of the circulating guided mode, and $r$ is the radius of the microtoroid. The group indexes of the fundamental TE and TM modes were extracted from the 2D finite difference eigenmode waveguide simulations (Supplementary Fig. 9). Based on Eq. (1), we chose the radius of the microtoroid to be 15 μm. This radius is large enough to provide a short free spectral range that allows capturing multiple resonances within the wavelength tuning range of the laser used for characterization, and to limit bend losses that would otherwise prevent the resonances to form. At a wavelength of 1550 nm, Eq. (1) predicts an FSR of 16 nm for the fundamental TM mode, which results in six or seven observable resonances in the wavelength range. The predicted FSR agrees well with the measured FSR (Supplementary Fig. 8c). The directional coupler (i.e., air gap) between the bus waveguide and the microtoroid was designed to have a width of 500 nm and a length of 23 μm. These dimensions ensured the coupling of all the modes between the waveguide and the toroid, and they were not selected to achieve critical coupling for any specific mode. Finally, the ends of the bus waveguide were directed orthogonal to each other to prevent stray

light from the input fiber to be reflected directly to the output fiber by the substrate surface (Fig. 3a, b).

## Optical characterization of the microtoroid resonator

The performance of the resonator was characterized by measuring its transmission spectrum between 1460 nm and 1580 nm. Light for the transmission experiments was produced by a linearly polarized tunable laser source with a fixed power of 1 mW (8164A, Agilent Technologies). The transmitted light was measured by a wavelength domain component analyzer (86082A, Agilent Technologies). Single-mode optical fibers were used to transfer light from the laser source to the resonator and back to the analyzer. To effectively couple light between the fibers and the ends of the bus waveguide, fibers with tapered tips (TSMJ-X-1550-9/125-0.25-7-5-26-2.1, AMS technologies) were used. The fiber tips had a working distance of 26 μm and a spot diameter of 5 μm and they were positioned using six linear piezo stages (AG-LS25, Newport). Two microscopes that imaged the positioning from above and from the side facilitated the alignment of the fiber tips to the ends of the bus waveguide. A fiber polarization controller (FPC030, Thorlabs) was used to control the direction of the linear polarization of the light coupled into the bus waveguide. This allowed us to preferably excite either TM or TE modes in the bus waveguide. The light transmitted through the resonator device was recorded in two components, with orthogonal polarizations, separated using a beam splitter (PBC1550SM, Thorlabs). Because the multiple modes supported by the bus waveguide had smaller mode diameters than the mode in the optical fibers, there was a significant mode mismatch in the coupling between the fibers and the bus waveguide. As a result, the modes excited in the waveguide were sensitive to the exact positioning of the fiber tips, and the ratio between excited modes in the bus waveguide depended on the fiber input position. When measuring the device transmission, we recorded spectra for multiple positions of the input fiber. The transmission measurements were first done for the as-printed resonator and repeated after annealing the resonator at temperatures of 150 °C, 300 °C, and 900 °C.

## Modeling and fitting of the transmission spectrum of the resonator

The measured resonance spectra of the 3D-printed silica glass microtoroid resonator were analyzed by fitting an all-pass, single-mode ring resonator model, to allow extraction of the FSR and quality factor of the resonator as a function of the annealing temperature. Before the fitting, the orthogonal polarization components of each spectrum were recombined. Each transmission spectrum corresponding to a specific fiber position and annealing temperature was fitted to the model function

$$T(\lambda) = \frac{a^2 - 2a\tau\cos\varphi(\lambda) + \tau^2}{1 - 2a\tau\cos\varphi(\lambda) + a^2\tau^2} \, , \qquad (2)$$

where $T$ is the light transmission at the output of the bus waveguide, $\lambda$ is the wavelength of the light, $a$ is the light transmission through the ring per revolution, $\tau$ is the light transmission in the bus waveguide through the directional-coupler region, and $\varphi(\lambda)$ is the accumulated phase per revolution of the guided mode circulating through the ring[49]. The function $\varphi(\lambda)$ is given as

$$\varphi(\lambda) = \frac{2\pi n_{eff}(\lambda)2\pi r}{\lambda} \, , \qquad (3)$$

where $n_{eff}$ is the effective index of the circulating guided mode and $r$ is the radius of the ring. The 3D-printed microtoroid had a radius of 15 μm, translating to a 15 μm ring radius in the ring resonator model in Eqs. (2) and (3). Because $n_{eff}$ was approximated as linearly varying with $\lambda$, fitting included the following four free parameters: $a$, $\tau$, $n_{eff}$ at

1550 nm, and the slope of $n_{eff}$ at 1550 nm. The fitting algorithm used was the genetic algorithm from Matlab's Global Optimization toolbox (Matlab R2019A). The parameter boundaries required for the algorithm were set based on the initial electromagnetic simulations shown in Supplementary Fig. 9. This global optimization algorithm was chosen over more conventional local, nonlinear curve-fitting algorithms to relax the need of initial estimates for the free variables. The local optimization algorithms that were tested often converged to local minima of the fitness function because the spectra included periodic resonances.

## Extraction of FSR and quality factor of the resonator

As discussed above, multiple spectra for each annealing temperature were collected using varying fiber positions, and all these spectra were fitted with the all-pass, single-mode ring resonator model. From these spectra, a single spectrum was chosen for each annealing temperature to be used for extracting the FSR and the spectrally averaged quality factor of the resonator at the said annealing temperature (Supplementary Fig. 8c, d). The selection of the spectra was based on the fitness of the all-pass, single-mode ring resonator model to the spectra. The spectra with the best fitness had the smallest contribution from secondary peaks, which in turn suggests they were the spectra with the strongest coupling to the fundamental TM and TE modes in the bus waveguide (Supplementary Fig. 8a, b). The FSR was obtained in two steps, where the fitted model first gave the wavelength-dependent effective index, which was in turn used to derive the group index $n_g$ in Eq. (1) to obtain the FSR. The obtained FSR agrees well with the expected value of 16 nm from design simulations, as the obtained effective index at 1550 nm is close to the expected value based on the MODE cross-section simulations. This confirms that our 3D-printed glass has an optical index close to that of fused silica since we used fused silica in our Lumerical simulations (SiO$_2$ (Glass) – Palik). The loaded (i.e., including the losses from the directional coupler) quality factor of the ring resonator was extracted from the fitted FSR, $a$, and $\tau$ using the analytical expression[49] for an all-pass ring resonator

$$Q = \frac{\pi n_g 2\pi r \sqrt{a\tau}}{\lambda_{res}(1 - a\tau)} = \frac{\pi \lambda_{res}\sqrt{a\tau}}{FSR(1 - a\tau)} = \frac{\lambda_{res}}{FWHM} \, . \qquad (4)$$

The quality factor $Q$ based on the fit over the whole wavelength range was used to obtain comparable values between the spectra because the quality factors of individual resonance peaks within a single spectrum had some variation, which can originate from the dispersion of the coupling coefficient $\tau$ at different wavelengths and from higher order modes in our multimode waveguide.

## Data availability

The data that supports the findings of this study are available in the article and supplementary information file and the source data are provided as a Source Data file. Source data are provided with this paper.

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

## Acknowledgements

We thank Cecilia Aronsson for substrate dicing and assistance in the cleanroom, Saumey Jain for assistance with AFM characterization, Kevin Kistermann for FIB preparation, Sebastian Zischke for the TEM characterization and investigations, Tanja Stimpel-Lindner for assistance with material analyses, and Dr. Fredrik Gustavsson from Swerim AB for the EDS characterization. This work has been funded by Swedish

Foundation for Strategic Research (SSF GMT14-0071 (G.S.) and SSF ID17-055 (J.N.)), Sweden Taiwan Research Projects 2019 (SSF STP19-0014 (K.B.G.)), Digitalization and Technology Research Center of the Bundeswehr under the project VITAL-SENSE (G.S.D. and O.H.) via German Recovery and Resilience Plan NextGenerationEU by the European Union, and European Union Horizon 2020 research and innovation program under the project Graphene Flagship Core 3 No. 881603 (G.S.D.) and SSLiP No. 101046693 (G.S.D.).

## Author contributions

P.H. and M.L. contributed equally. M.L., C.E.H., and K.B.G. proposed the concept. P.H., M.L., G.S., K.B.G., and F.N. conceived and designed the experiments. P.H. fabricated the structures on silica-glass substrates. O.H. and G.S.D. performed Raman and photoluminescence spectroscopy characterizations. J.M. performed TEM-based characterizations. P.H. and M.L. performed microscopy characterizations and annealing experiments. M.L. analyzed the Raman, photoluminescence, and shrinkage data, and P.H. analyzed the TEM data, with contributions from all authors. P.E., P.H., and K.B.G. designed and characterized the microtoroid resonator, and analyzed the related data. P.H., P.E., L.L., K.B.G., and F.N. conceived and designed the printing experiments on the tips of optical fibers, and L.L. performed the experiments. J.N. and P.H. performed the nanoindentation experiments and analyzed the data. All authors discussed the results and contributed to the manuscript writing.

## Funding

## Competing interests

A patent application (US 17/171,587) covering the methods and the optical applications in this work has been filed, with M.L., P.H., P.E., L.L., C.E.H., G.S., K.B.G., and F.N. as inventors and applicants. The remaining authors declare no competing interests.
