## [Peer Review File · Nature Communications]

Reviewer comments, first round

Reviewer #1 (Remarks to the Author):

This paper demonstrates 3D printing of silica glass that is solid and optically transparent as printed and features sub-micrometer resolution. Hydrogen silsesquioxane (HSQ) without any additives is selectively crosslinked into silica glass in 3D by exposure to sub-picosecond laser pulses without the need of a sintering step which is fairly innovative. Furthermore, compared to fused silica glass, the as-printed material features a higher ratio of 4-membered silicon-oxygen rings in the network. This printing approach enables promising applications in important fields of science and technology, such as photonics, medicine, and quantum-optics. Overall the study is very interesting. However, a few points needed be addressed before it can be considered for publication.

1. The resolution and quality of Figure 1 in page 6 could be further improved.
2. Most figure statement covers too many characters, and the discussion sentences could be brought to the experiment or the text part.
3. Add some measurements on the mechanical properties of the printed material can further enrich the content of the paper.

Reviewer #2 (Remarks to the Author):

The authors describe a 3D printing technique to create 3D silica glass structures with sub-micrometer resolution without an annealing step. The result is obtained using sub-picosecond laser pulses with a wavelength of 1040 nm on commercial resist HSQ. The as printed glass is optically transparent and presents a broad photoluminescence peak. To obtain a material which is indistinguishable from the commercial silica glass, a further annealing at 900°C is necessary, which causes low shrinkage of the structure. To demonstrate the technique, three different optic components have been fabricated.

The work strongly pushes forward the research conducted both on 3D printing and on silica glass micro/nanofabrication. Even if 3D printed silica structures have been already obtained, they require the use of colloidal silica nanoparticles and therefore an annealing step at 1100-1300°C that provokes a high shrinkage of minimum 15% (Wen et al, Nature Materials, 2021). Even if other groups have already demonstrated a double layered grid with femtosecond laser writing on HSQ at 780 nm (Jin et al, Nature Communications, 2022), the present research introduces fully 3D structures, and, even more important, demonstrates the use of HSQ as a precursor of silica glass. The potentiality for optics, photonics, and sensing is evident.

The absence of an annealing step allows to create optic devices compatible with materials that cannot reach high temperatures like polymers. Moreover, the possible further annealing step at 900°C that eliminates the photoluminescence of the structures and makes them indistinguishable from commercial silica glass with a shrinkage of only 6% is noteworthy. Finally, the possibility to introduce Si nanocrystals in the printed silica glass widens its possible applications.

The conclusions are supported by the described work, the methodology is sound and the details consent to reproduce the results.

However, some comments and clarifications would be useful to improve the readability of the manuscript. In the following I enter into details.

Fig 2b page 9. An optical microscope top view picture of the ring structure without the glass plate with four legs would be useful as a reference. If there is no space in fig 2, it can be added to the supplementary information.

Effect of annealing on the 3D-printed structures

Page 10 from line 217 to line 219. Why temperature of 1200 °C was selected to test the delamination? The shrinkage was measured at 900°C, while the structure reduction at 1200°C is due to reflow and smoothening of the test structures and not to material shrinkage. A comment should be added.

3D-printed optical microtoroid resonator in silica glass

Page 12 line 260. How is the performance of the resonator defined? Is it in term of FSR and Q? In this case, a percentage indicating the fluctuation of Q value with the temperature could be useful. In addition, Supplementary figure 6c is not easily readable. If both FSR and Q are put in the same graph, a legend should be added explaining that circles are relative to FSR and triangles to Q (if I understood correctly). Then, the scale of Q should be changed as it does not easily allow to read the variation with temperature. The value of Q for TE input light looks much lower for treatment at 150° and 300° C with respect to the one at 900°C, the one for TM looks quite higher at 300°C with respect to the one at 900°C, and this would be in contrast with the statement of the stability of the performance across all annealing temperatures.

Methods. Annealing of 3D-printed samples

The brand of the oven should be added

Reviewer #3 (Remarks to the Author):

Manuscript titled "Three-dimensional printing of silica glass with sub-micrometer resolution," by P-Huang et al. is a very interesting paper describing the fabrication and characterization of 3D parts with sub-micron features obtained by sub-picosecond laser pulse. The topic is certainly innovative and of interest to the readers. The work has been carried out carefully and competently, and the authors tried to address all the different aspects related to their material and micro-devices. I have no problems with the data and the work and how it has been described and discussed in the manuscript.

However, I have some reservations at calling "silica glass" a material constituted by crosslinked HSQ. Besides the difference in the amount of 3 and 4-membered rings (before annealing) with respect to fused silica, there is certainly a much larger amount of Si-H bonds in the as-printed material versus in silica glass. For other precursor-derived silica materials, for instance, it is well known that well crosslinked silica films obtained from fully hydrolyzed and condensed TEOS are different from silica glass in terms of hardness and density (true, in this case we have the solvent evaporation leaving behind capillary voids). Nevertheless, I do wonder if the density and hardness of as-printed HSQ samples are, indeed, similar to those of fused silica.

I believe that, before publication, the authors should at least perform nano-hardness tests on the as printed samples, and compare the results with those of a fused silica sample. Ideally, if they can print parts large enough, they could also perform density measurements.

On a minor note:

1) in the introduction, the authors forgot to mention the possibility of fabricating silica glass structures using a hybrid approach (UV-assisted DIW), employing colloidal silica and TEOS as silica precursor (see: <https://doi.org/10.1016/j.addma.2022.102727>). This work should also be quoted and commented in the introduction;

2) in the introduction, the authors state: "This is because any substrate material involved in the printing process must withstand the thermal treatment, which essentially eliminates most materials of interest". However, there is not necessarily a substrate material when using DLP, SLA and DIW (it depends on the form of the object how you attach it to the platform, for DLP and SLA, and the reported work by Dilla-Spears et al. shows the fabrication of self-standing lenses, while the work by De Marzi et al. reports the manufacturing of bulk pieces with an architected morphology). This sentence should therefore be corrected;

3) I do not see reported anywhere the source for the HSQ material; this should be clearly indicated in the manuscript.

The manuscript can be published after minor revisions.

Point-by-Point Response to the Comments of the Reviewers

We thank [redacted] the reviewers for the constructive and insightful comments, which helped us to substantially improve our paper. To address all comments of the reviewers, we have performed new experiments, and revised the manuscript and included additional data, analysis, discussions, and references. In the following, we provide detailed responses to all comments of the reviewers.

Reviewer #1:

Reviewer's comment:

- 1. This paper demonstrates 3D printing of silica glass that is solid and optically transparent as printed and features sub-micrometer resolution. Hydrogen silsesquioxane (HSQ) without any additives is selectively crosslinked into silica glass in 3D by exposure to sub-picosecond laser pulses without the need of a sintering step which is fairly innovative. Furthermore, compared to fused silica glass, the as-printed material features a higher ratio of 4-membered silicon-oxygen rings in the network. This printing approach enables promising applications in important fields of science and technology, such as photonics, medicine, and quantum-optics. Overall the study is very interesting.*

Our response:

We thank the reviewer for the very positive assessment of our work.

Reviewer's comment:

2. However, a few points needed be addressed before it can be considered for publication.

1. The resolution and quality of Figure 1 in page 6 could be further improved.

Our response:

We thank the reviewer for this comment. We agree that the quality of some SEM images in Figure 1 requires further improvements.

To address this comment in the revised manuscript, we have updated Fig. 1d by a newly taken SEM image of the same goblet with an increased resolution and reduced charging effects. In addition, we have replaced Fig. 1e by an SEM image taken from a newly 3D-printed spiral with a design that more clearly demonstrates the stability of our 3D-printed structures:

Revised Manuscript, Page 7, Line 115-128:

Fig. 1. 3D printing of silica-glass microstructures by direct laser writing. **a**, Preparation of HSQ by drop casting. **b**, Direct writing in the HSQ using the focus of the sub-picosecond laser beam. **c**, Development of the 3D-printed structures by dissolving the unexposed HSQ. The inset SEM image shows a 3D-printed KTH logo (scale bar, 1 μm). **d**, **e**, SEM images of 3D-printed structures: a glass goblet and a conical spiral (scale bars, 1 μm). **f**, **g**, SEM images of an array of 500 nm diameter needles and of a

pillar with a longitudinal twist (scale bars, 1 μm). **h, i**, SEM images of a cantilever that is 750 nm wide and 540 nm thick taken from a 45-degree angle. The roughness on the top surface originates from a deposited gold layer for reducing charging effects during imaging (scale bars, 1 μm). **j**, Scanning TEM image of an as-printed silica glass structure. The dark region on the bottom-left corner is the metal layer deposited during sample preparation, and the interface between the glass and metal is marked with a dashed line (scale bar, 50 nm). **k**, Electron diffraction pattern of the TEM sample. The pattern consists of concentric rings, showing that the printed material is amorphous (scale bar, 5 nm^{-1}).

Reviewer's comment:

3. 2. Most figure statement covers too many characters, and the discussion sentences could be brought to the experiment or the text part.

Our response:

We thank the reviewer for this comment. We agree that the figure captions should be shortened to improve their accessibility and clarity.

To address this comment in the revised manuscript, we have revised the captions of Fig. 1, 2, and 3, and moved the detailed discussions to the related text passages in the main text. The captions and the related text passages now read:

Revised Manuscript, Page 7, Line 115-128:

Fig. 1. 3D printing of silica-glass microstructures by direct laser writing. **a**, Preparation of HSQ by drop casting. **b**, Direct writing in the HSQ using the focus of the sub-picosecond laser beam. **c**, Development of the 3D-printed structures by dissolving the unexposed HSQ. The inset SEM image shows a 3D-printed KTH logo (scale bar, 1 μm). **d**, **e**, SEM images of 3D-printed structures: a glass goblet and a conical spiral (scale bars, 1 μm). **f**, **g**, SEM images of an array of 500 nm diameter needles and of a pillar with a longitudinal twist (scale bars, 1 μm). **h**, **i**, SEM images of a cantilever that is 750 nm wide and 540 nm thick taken from a 45-degree angle. The roughness on the top

surface originates from a deposited gold layer for reducing charging effects during imaging (scale bars, 1 μm). **j**, Scanning TEM image of an as-printed silica glass structure. The dark region on the bottom-left corner is the metal layer deposited during sample preparation, and the interface between the glass and metal is marked with a dashed line (scale bar, 50 nm). **k**, Electron diffraction pattern of the TEM sample. The pattern consists of concentric rings, showing that the printed material is amorphous (scale bar, 5 nm^{-1}).

Revised Manuscript, Page 10, Line 176-191:

Fig. 2. Characterization of the 3D-printed glass and effects of annealing. **a**, Raman spectra of 3D-printed silica glass structures without annealing (i.e., as-printed) and after annealing at different temperatures, and Raman spectrum of a commercial silica glass substrate for comparison after the removal of non-Raman background signals and normalization. The positions of the Raman peaks are summarized in Supplementary Table 2. An additional spectrum without the background removal of the sample annealed at 300 °C is plotted overlapping with the spectrum after the background removal, to illustrate the observed sloped background. **b**, Relative linear shrinkage of the 3D-printed silica glass test structures, shown in Supplementary Fig. 5b, after annealing at different temperatures. The error bars show the largest measured variation from the mean value or the measurement uncertainty, whichever was larger. **c**, SEM image of a 3D-printed silica glass plate suspended on four legs with a ring structure underneath. Below the SEM image is an optical-microscope top-view of the ring structure imaged through the suspended plate. **d**, **e**, **f**, The same structure as in **c** after further annealing at 900 °C for one hour, 1200 °C for one hour, and 1200 °C for two hours, respectively. (Scale bars, 1 μm)

Revised Manuscript, Page 16, Line 313-328:

Fig. 3. 3D-printed optical demonstrators and their characterization. **a**, SEM image of a 3D-printed photonic microtoroid resonator with a bus waveguide without annealing (i.e., as-printed). **b**, SEM top view of the same device as in **a**, superimposed with the simulated vertical component of the electrical field at resonance when injecting TM_{00} mode into the bus waveguide (Supplementary Fig. 9). **c**, Measured transmission spectra of the resonator in **a** without annealing and after annealing at 900 °C when injecting dominantly TM_{00} mode, normalized to their respective maximum values. Solid lines show the results of a resonator model fitted to the measured transmission spectra. **d**, **e**, Optical microscope image with top view of two cubes exposed by laser pulses only at its shell and through the entire volume, respectively, after annealing at 1200 °C, superimposed with the corresponding photoluminescence images. **f**, Photoluminescence spectra of the core of the

shell-exposed cube in **d**, the fully exposed cube in **e**, and the silica glass substrate. The intensity of the displayed spectra of the fully exposed cube and the substrate is scaled up by a factor of 20 for improved clarity. **g**, SEM image of a suspended glass plate 3D-printed onto the tip of an optical fiber. (Scale bars, 10 μm)

Revised Manuscript, Page 13, Line 260-267:

“We characterized the 3D-printed resonator by measuring its transmission spectra in the optical telecommunication S, C, and L bands between 1460 nm and 1580 nm when injecting the fundamental transverse magnetic (TM_{00}) and electric (TE_{00}) mode into the bus waveguide. To monitor any possible effects of annealing on the functionality of the resonator, we characterized the resonator as printed, as well as after annealing at 150 $^{\circ}\text{C}$, 300 $^{\circ}\text{C}$, and 900 $^{\circ}\text{C}$, respectively. The spectra measured before annealing and after annealing at 900 $^{\circ}\text{C}$ with the TM_{00} mode injection are shown in Fig. 3c, and the spectra for other annealing temperatures and TE_{00} mode injection are shown in Supplementary Fig. 8a, b.”

Revised Manuscript, Page 15, Line 302-308:

“After annealing of the cubes at 1200 $^{\circ}\text{C}$ in air, a strong photoluminescence peak, centered at a wavelength of 670 nm (1.85 eV), and several Raman features at below 4000 cm^{-1} that do not belong to silica glass were observed in the volume of the unexposed HSQ (Fig. 3f and Supplementary Fig. 12). The photoluminescence peak indicates the presence of silicon nanocrystals, while the laser-exposed shell, as well as the fully laser-exposed cube, showed little to no photoluminescence (Fig. 3d-f).”

Reviewer's comment:

- 4. 3. *Add some measurements on the mechanical properties of the printed material can further enrich the content of the paper.***

Our response:

We thank the reviewer for this important comment, which inspired us to further investigate the mechanical properties of the 3D-printed glass. Therefore, we have performed new nanoindentation experiments and analyses based on the Oliver-Pharr method¹ to obtain hardness and reduced elastic modulus for three different sample types. The first sample type consists of a microplate with a footprint of 20 μm x 20 μm and a thickness of about 2 μm , 3D-printed on a fused-silica substrate. The fused-silica substrates that we used here, and throughout the work, are JGS2 optical-grade fused quartz purchased from MicroChemicals. The second sample type consists only of a flat JGS2 fused-silica substrate for reference. Finally, to consider potential effects of the geometry of the plates on the indentation measurement results, we fabricated a third sample type by engraving microplates of the same dimensions as the 3D-printed plates into the same JGS2 fused-silica substrate using laser ablation. For the indentation measurements, we used two specimens of each sample type, one of the specimens was annealed at 900 °C for one hour, and the other one was not.

We have summarized the results of the nanoindentation experiments in Supplementary Table 3, and one representative loading-unloading curve for each sample type is shown in Supplementary Fig. 7. The results and the curves are also included below. The measured hardness and reduced elastic modulus of the flat JGS2 fused silica substrates before and after annealing at 900 °C were similar for both specimen which are about 9.5 GPa and 70 GPa, respectively. These values are in line with the expected values of fused silica². However, the measured hardness and reduced elastic modulus of laser-ablated fused-silica microplates before and after annealing were about 7.8 GPa and 65 GPa which are slightly lower than those of flat substrates. This shows that the geometry and dimensions of the microplates must be considered when interpreting the nanoindentation measurement results. Therefore, we will primarily compare the results of the 3D-printed microplates to those of laser-ablated fused-silica microplates in the following discussion since the laser-ablated samples are the most relevant reference samples because they are similar in geometry and dimensions to our 3D printed glass plates.

The measured hardness and reduced elastic modulus of the 3D-printed microplates were 2.4 ± 0.2 GPa and 40 ± 2 GPa, respectively, as printed. After annealing at $900\text{ }^{\circ}\text{C}$ for 1 h, the hardness and reduced elastic modulus values reached 7.7 ± 0.6 GPa and 75 ± 2 GPa, respectively, i.e., similar values as the fused silica samples of the same geometry. The somewhat lower values of our as-printed glass are expected since we have observed residual hydrogen, hydroxyl groups, and water content in its Raman spectrum. Also, we observed that the as-printed glass shrinks by a small amount (6.1 %) upon annealing. Interestingly, the hardness and reduced elastic modulus of our 3D-printed glass reach the values of the reference fused-silica microplates after annealing at $900\text{ }^{\circ}\text{C}$. This suggests that the residual components in our 3D-printed glass were completely removed, and that the glass was fully densified at an annealing temperature of $900\text{ }^{\circ}\text{C}$. This agrees with our observation that the Raman spectrum of the 3D-printed glass after annealing is indistinguishable from that of fused silica. This is also supported by our observation that the material shrinkage reached the ultimate value of 6.1% after annealing at $900\text{ }^{\circ}\text{C}$. Along with our EDS data and the fact that no pores were observed by TEM in the as-printed glass, down to the scale of a few nanometers, we conclude that the as-printed glass is low-density silica glass with a residual amount of hydrogen, hydroxyl groups, and water, and that annealing at $900\text{ }^{\circ}\text{C}$ transforms it into fully densified silica glass that is indistinguishable from fused silica.

In the revised manuscript and supplementary information, we have included the new experimental results, analysis and discussion that we performed in response to this comment of the reviewer. Specifically, we have included hardness and reduced elastic modulus measured from the 3D-printed glass and references. We have also analyzed and discussed these results in depth in the related text passages, which now read:

Revised Manuscript, Page 4 and 5, Line 78-85:

“We show by Raman, energy-dispersive, and photoluminescence spectroscopy that the as-printed material is silica glass but, compared to fused silica glass, features a higher ratio of 4-membered silicon-oxygen rings in the network resulting from sub-picosecond laser exposure, photoluminescence, residual hydrogenated and hydroxyl species, and trace amounts of organic residuals. These features and residuals can be removed by a $900\text{ }^{\circ}\text{C}$ annealing step, resulting in a low shrinkage of 6.1% of the 3D-printed structures and an increase of the hardness and reduced elastic modulus of the 3D-printed silica glass to values expected for fused silica glass.”

Revised Manuscript, Page 12, Line 230-249:

“The mechanical properties of the 3D printed silica glass are important for its use in many applications, including nanoelectromechanical systems (NEMS). With this in mind, we characterized the hardness and reduced elastic modulus of the 3D-printed silica glass by performing nanoindentation measurements on 3D-printed microplates with a footprint of 20 μm by 20 μm and a thickness of about 2 μm . The measured hardness and reduced elastic modulus of the as-printed silica glass were 2.4 ± 0.2 GPa and 40 ± 2 GPa, respectively. After annealing at 900 $^{\circ}\text{C}$ for 1 h, the hardness and reduced elastic modulus increased to 7.7 ± 0.6 GPa and 75 ± 2 GPa, respectively, which are almost identical to the values we measured of reference samples consisting of fused-silica microplates of the same geometry (Supplementary Table 3 and Supplementary Fig. 7). The differences in the hardness and reduced elastic modulus between the as-printed glass and the fused silica reference sample are expected since we have observed residual hydrogen, hydroxyl groups, and water content in the Raman spectrum of the as-printed glass, and since the as-printed glass shrinks by up to (6.1 ± 0.8) % upon annealing (Fig. 2a, b). Our observation that the hardness and reduced elastic modulus of the 3D-printed glass after annealing at 900 $^{\circ}\text{C}$ reached the values of the fused-silica microplate reference samples, suggests that the residual components in our 3D-printed glass were completely removed, and that the glass was fully densified at an annealing temperature of 900 $^{\circ}\text{C}$. This is also consistent with our observations that the Raman spectrum of the 3D-printed glass after annealing is indistinguishable from that of fused silica, and that the material shrinkage reached its ultimate value of 6.1% after annealing at 900 $^{\circ}\text{C}$.”

Revised Manuscript, Page 23, Line 480-496:

“**Nanoindentation characterization.** Three types of samples were characterized. The first sample type consists of a microplate with a footprint of 20 μm x 20 μm and a thickness of about 2 μm , 3D-printed on a fused-silica substrate. The second sample type consists only of a flat fused-silica substrate for reference. Finally, to consider potential effects of the geometry of the plates on the indentation measurement results, we fabricated a third sample type by engraving microplates of the same dimensions as the 3D-printed plates into a fused-silica substrate using laser ablation. In all experiments, the fused-silica substrates were JGS2 optical-grade fused quartz purchased from MicroChemicals GmbH.

For the indentation measurements, we used two specimens of each of the three sample types, one of the specimens was annealed at 900 °C for one hour, and the other specimen was not annealed. A nanoindenter (Hysitron TI 950 Triboindenter, Bruker) was used to measure the hardness and reduced elastic modulus of the samples via the Oliver–Pharr method¹. The indenter was equipped with a Berkovich tip and was calibrated using a standard quartz reference sample with a hardness and elastic modulus of 9.3 and 69.6 GPa, respectively. All measurements were run in the load-controlled mode with set maximum loads that yielded indents that are sufficiently deep to avoid tip radius influence and do not exceed 200 nm, that is, not more than 10% of the thickness of the microplates, to avoid contributions from the substrate. On each sample, three measurements were performed.”

Revised Supplementary Information, Page 4:

Supplementary Table 3. Summary of nanoindentation characterization results.

Measurement results of our 3D-printed microplates, along with reference samples consisting of laser-ablated fused-silica microplates and flat fused-silica substrates. All sample types were measured without, and with annealing at 900 °C. For each sample, three measurements were performed. The measured hardness and reduced elastic modulus values are in the format of average \pm standard deviation of the three measurements of each sample.

Sample	Hardness (GPa)	Reduced elastic modulus (GPa)
3D-printed microplate without annealing	2.4 \pm 0.2	40 \pm 2
3D-printed microplate annealed at 900 °C	7.7 \pm 0.6	75 \pm 2
Laser-ablated fused-silica microplate	7.4 \pm 0.1	63.8 \pm 0.6
Laser-ablated fused-silica microplate annealed at 900 °C	8.1 \pm 0.5	66 \pm 1
Flat fused-silica substrate	9.44 \pm 0.05	69.7 \pm 0.7
Flat fused-silica substrate annealed at 900 °C	9.62 \pm 0.03	71.0 \pm 0.1

Revised Supplementary Information, Page 11:

Supplementary Fig. 7. Nanoindentation characterization of the 3D-printed glass and the fused silica glass reference samples. Loading-unloading curves of the samples listed in Supplementary Table 3. One of the measured three curve from each sample is plotted to demonstrate the behavior of the samples in the indentation experiments.

Reviewer #2:

Reviewer's comment:

1. *The authors describe a 3D printing technique to create 3D silica glass structures with sub-micrometer resolution without an annealing step. The result is obtained using sub-picosecond laser pulses with a wavelength of 1040 nm on commercial resist HSQ. The as printed glass is optically transparent and presents a broad photoluminescence peak. To obtain a material which is indistinguishable from the commercial silica glass, a further annealing at 900°C is necessary, which causes low shrinkage of the structure. To demonstrate the technique, three different optic components have been fabricated.*

The work strongly pushes forward the research conducted both on 3D printing and on silica glass micro/nanofabrication. Even if 3D printed silica structures have been already obtained, they require the use of colloidal silica nanoparticles and therefore an annealing step at 1100-1300 °C that provokes a high shrinkage of minimum 15% (Wen et al, Nature Materials, 2021). Even if other groups have already demonstrated a double layered grid with femtosecond laser writing on HSQ at 780 nm (Jin et al, Nature Communications, 2022), the present research introduces fully 3D structures, and, even more important, demonstrates the use of HSQ as a precursor of silica glass. The potentiality for optics, photonics, and sensing is evident.

The absence of an annealing step allows to create optic devices compatible with materials that cannot reach high temperatures like polymers. Moreover, the possible further annealing step at 900°C that eliminates the photoluminescence of the structures and makes them indistinguishable from commercial silica glass with a shrinkage of only 6% is noteworthy. Finally, the possibility to introduce Si nanocrystals in the printed silica glass widens its possible applications.

The conclusions are supported by the described work, the methodology is sound and the details consent to reproduce the results.

Our response:

We thank the reviewer for the very positive assessment of our work in terms of its contribution to the advancement in 3D printing and glass micro/nanofabrication, its potential for optical and

sensing applications, and the methodology.

Reviewer's comment:

- 2. However, some comments and clarifications would be useful to improve the readability of the manuscript. In the following I enter into details.*

Fig 2b page 9. An optical microscope top view picture of the ring structure without the glass plate with four legs would be useful as a reference. If there is no space in fig 2, it can be added to the supplementary information.

Our response:

We thank the reviewer for this excellent suggestion. We have printed a structure that consists of a ring and four legs which are identical to the corresponding parts of the structure shown in Fig. 2b without a suspended plate above the ring. We have taken optical-microscope top view and SEM images of the structure without a suspended plate. By comparing Fig. 2c and Supplementary Fig. 4, we observed that the ring can be clearly seen through the suspended plate, as is the case for the ring without a suspended plate. The view of the ring through the suspended plate was slightly distorted compared to the view of the ring without a suspended plate. This distortion originates from the surface topography of the suspended plate. In the case of Fig. 2e and 2f, which were images taken after the sample was smoothed and reflowed at 1200 °C, the ring remained as visible as the new image taken from the ring without a suspended plate, and the distortion observed in Fig. 2c and 2d was minimized due to the smoothing of the suspended plate. In summary, we demonstrated that the as-printed glass is optically transparent to such an extent that a pattern below a 3 μm thick suspended glass plate was clearly visible when viewing it through the plate, while the view of the pattern was slightly distorted due to the topography of the plate and the distortion can be minimized by annealing.

In the revised manuscript and supplementary information, we have included the optical-microscope and SEM images of the new structure, and included a discussion of the images taken with and without a suspended plate:

Revised Manuscript, Page 10 and 11, Line 195-204:

“To show that our 3D-printed glass features a level of optical transparency that is relevant for micro-optical components, we 3D-printed structures that consist of a 3 μm thick suspended glass plate that is placed above a ring on the substrate. We imaged the ring through the plate using optical microscopy, both before and after annealing (Fig. 2c, d). In addition, as reference we 3D-printed the same ring structure without the suspended

plate (Supplementary Fig. 4). By comparing Fig. 2c and Supplementary Fig. 4, we observed that the ring can be clearly seen through the suspended plate, as is the case for the ring without a suspended plate. The view of the ring through the suspended plate was slightly distorted compared to the view of the ring without a suspended plate. This distortion originates from the surface topography of the suspended plate.”

Revised Supplementary Information, Page 8:

Supplementary Fig. 4. Reference images of the 3D-printed glass ring without a suspended glass plate above the ring structure. SEM image and optical microscope top-view image of the 3D-printed silica glass ring without a suspended plate, as a reference for the images of the ring with a suspended plate (Fig. 2c-f). To include the potential effects of the supporting legs of the plate in Fig. 2c-f on the view of the ring, identical legs (but without plate) were printed around the reference ring (scale bar, 5 μm).

Reviewer's comment:

3. Effect of annealing on the 3D-printed structures

Page 10 from line 217 to line 219. Why temperature of 1200 °C was selected to test the delamination? The shrinkage was measured at 900°C, while the structure reduction at 1200°C is due to reflow and smoothening of the test structures and not to material shrinkage. A comment should be added.

Our response:

We thank the reviewer for pointing this out and for the opportunity to clarify the discussion about the delamination experiments of our 3D-printed structures. We have conducted the delamination experiments at both 900 °C and 1200 °C by taking optical-microscope and SEM images for a structure with four attachment points to the substrate after annealing at 900 °C and 1200 °C, respectively, as shown in Fig. 2d-f. For 900 °C, it is the temperature that the Raman spectrum of the 3D-printed glass became indistinguishable from that of fused silica, so it is important to confirm that the material shrinkage of the 3D-printed glass structures caused by annealing at 900 °C does not lead to the delamination of the structure from the substrate. As for 1200 °C, it is the glass-transition temperature for fused silica at which reflow and surface smoothening of fused silica become significant, so it is the highest temperature that the 3D-printed glass structures would undergo and be expected to keep their shapes to some extent. We observed no delamination of the structure from the substrate after annealing the sample at either temperature, which means that neither material shrinkage nor reflow and surface smoothening caused delamination of the 3D printed structures. The delamination experiments at 1200 °C for 2 hours was emphasized in the discussion because this was the most extreme condition that we have evaluated. We agree with the reviewer that the experiments at 1200 °C does not demonstrate the effects of material shrinkage.

In the revised manuscript, we have modified the related text passage to clarify that the delamination experiments have been conducted at both 900 °C and 1200 °C, which now reads:

Revised Manuscript, Page 11, Line 217-225:

“In contrast to these methods, the low shrinkage of our 3D-printed silica glass enables preservation of the shape of the 3D-printed structures and avoids delamination of the structures from the substrate. To demonstrate this important feature, we printed a suspended plate with four attachment points that were spaced by 10 μm to the substrate and annealed the structure at 900 °C for one hour (Fig. 2d). We observed no cracks in the

structure or delamination of the attachment points from the substrate after annealing with the associated material shrinkage of the structure. Further annealing of the 3D printed glass at 1200 °C caused reflow and surface smoothing of the glass structures. As discussed above, the reflow of the glass does not represent linear material shrinkage. Even in this extreme case, we did not observe cracks or delamination in our 3D printed glass structures after annealing at 1200 °C for up to two hours (Fig. 2b, e, f).”

Reviewer's comment:

4. 3D-printed optical microtoroid resonator in silica glass

Page 12 line 260. How is the performance of the resonator defined? Is it in term of FSR and Q? In this case, a percentage indicating the fluctuation of Q value with the temperature could be useful.

In addition, Supplementary figure 6c is not easily readable. If both FSR and Q are put in the same graph, a legend should be added explaining that circles are relative to FSR and triangles to Q (if I understood correctly). Then, the scale of Q should be changed as it does not easily allow to read the variation with temperature. The value of Q for TE input light looks much lower for treatment at 150 ° and 300 °C with respect to the one at 900 °C, the one for TM looks quite higher at 300 °C with respect to the one at 900 °C, and this would be in contrast with the statement of the stability of the performance across all annealing temperatures.

Our response:

We thank the reviewer for this important comment. We agree that the sentence (Original Manuscript, Page 12, Line 260) about the performance of the 3D-printed optical microtoroid resonator after annealing at different temperatures should be further clarified and that the presentation of Supplementary Fig. 6c in Original Supplementary Information should be improved. The essential information that we were trying to convey is that the resonator remains functional across all evaluated annealing temperatures up to 900 °C.

As background, we note that an optical resonator is a generic component commonly used as a filter or a sensor. For such applications, the separation between adjacent resonances (FSR) and their sharpness (Q) are indeed the essential characteristics. However, FSR and Q are not universal performance metrics in the sense that bigger is always better. In general, FSR and Q are adapted to the application at hand by the design of the resonator.

Since both FSR and Q are very sensitive to changes in geometry and material properties, monitoring them after each annealing step is a good way to assess the stability of the printed material and geometry. In our annealing experiment, we observe some variation of FSR and Q, but within a span that would be acceptable for most applications, since temperature differences during operation of more than 150 °C are rare in practice.

The slight increase of FSR with annealing is consistent with the material shrinkage observed by other means. The observed variation of Q is in our opinion within the experimental error. The resonator supports multiple modes and coupling to only the fundamental mode, for the most exact evaluation of Q, proved challenging. A slight variation in the position or the angle of the excitation laser relative to the facet of the bus waveguide can lead to a significant variation in the distribution of the input optical power in the different modes. Thus, each time we couple after annealing we excite higher order modes to a variable degree, as evident by the slight “shoulders” in the otherwise periodic resonance spectrum. These “shoulders” affect the fitting of the model needed to extract the Q. The evaluation of FSR is less sensitive to this effect, since that only requires us to locate the minima of the spectrum.

In the revised manuscript, we have improved the presentation of the data in Supplementary Fig. 6c in Original Supplementary Information by separating it into two figures which are now in Supplementary Fig. 8 in Revised Supplementary Information. Furthermore, we modified the text passages on the variation of Q with annealing, to clarify the discussion of the performance of the 3D-printed microtoroid resonator. This part now reads:

Revised Manuscript, Page 14, Line 284-293:

“The apparent variations in the quality factor of peaks in one spectrum and across spectra measured after annealing the resonator at different temperatures were due to the possibility of the resonator to support multiple modes (Supplementary Fig. 9). The evaluation of the quality factor of a peak is sensitive to the shape of the peak which depends on the distribution of optical power in different modes. The optical power distribution is in turn sensitive to changes in the coupling condition between the optical fibers and the waveguide. Thus, the resonator was functional and its FSR was stable and predictable across all evaluated annealing temperatures without significant effects of annealing on its quality factor, demonstrating that the 3D-printed silica glass can be used for photonic and optical micro-devices, both with and without a subsequent annealing step.”

Supplementary Fig. 8. Transmission, free spectral range (FSR), and quality factor of the microtoroid resonator. **a, b**, Measured transmission spectra of the microtoroid resonator using TM and TE input light, respectively. The transmission spectra were measured for the as-printed resonator and after annealing at different temperatures. The measured spectra were fitted with an all-pass, single-mode ring resonator model, shown as grey lines. **c, d**, The FSR and the quality factor of the resonator after annealing at different temperatures, respectively. The presented values were extracted from the fitted transmission spectra shown in **a** and **b**. The FSR is shown for a wavelength of 1550 nm while the quality factor corresponds to an average value over the wavelength range.

Reviewer's comment:

5. *Methods. Annealing of 3D-printed samples*

The brand of the oven should be added.

Our response:

We thank the reviewer for pointing this out.

In response to this comment, we have added the brand and model of the oven we used for annealing in this work, which now reads:

Revised Manuscript, Page 20, Line 420-421:

“For the 3D printed glass samples that were annealed, annealing was done in an oven (Metallwarenfabrik 51/s, Conrad Naber, Germany) in an air atmosphere.”

Reviewer #3:

Reviewer's comment:

1. *Manuscript titled "Three-dimensional printing of silica glass with sub-micrometer resolution," by P- Huang et al. is a very interesting paper describing the fabrication and characterization of 3D parts with sub-micron features obtained by sub-picosecond laser pulse. The topic is certainly innovative and of interest to the readers. The work has been carried out carefully and competently, and the authors tried to address all the different aspects related to their material and micro-devices.*

I have no problems with the data and the work and how it has been described and discussed in the manuscript.

Our response:

We thank the reviewer for the positive assessments of the novelty and the presentation of our manuscript and the experiments we did to investigate the printed material and the 3D-printed micro-devices.

Reviewer's comment:

2. *However, I have some reservations at calling "silica glass" a material constituted by crosslinked HSQ. Besides the difference in the amount of 3 and 4-membered rings (before annealing) with respect to fused silica, there is certainly a much larger amount of Si-H bonds in the as-printed material versus in silica glass. For other precursor-derived silica materials, for instance, it is well known that well crosslinked silica films obtained from fully hydrolyzed and condensed TEOS are different from silica glass in terms of hardness and density (true, in this case we have the solvent evaporation leaving behind capillary voids). Nevertheless, I do wonder if the density and hardness of as-printed HSQ samples are, indeed, similar to those of fused silica.*

I believe that, before publication, the authors should at least perform nano-hardness tests on the as printed samples, and compare the results with those of a fused silica sample. Ideally, if they can print parts large enough, they could also perform density measurements.

Our response:

We thank the reviewer for this insightful comment and the opportunity to clarify this important point. We agree that Si-H bonds were not completely removed in the as-printed glass since the Si-H peak at 2260 cm^{-1} was present in the Raman spectrum of the as-printed glass. However, the intensity of the Si-H peak was much smaller than that of the signature peaks of silica glass in the range from 180 cm^{-1} to 900 cm^{-1} . This is in strong contrast to the Raman spectrum of HSQ before the sub-picosecond laser exposure where the Si-H peak dominates over all other peaks³. This indicates that the amount of Si-H bonds in HSQ has been significantly decreased by the sub-picosecond laser exposure. This is supported by our further observation that a structure containing both exposed and unexposed HSQ only showed significant photoluminescence in the unexposed volume after annealing at $1200\text{ }^{\circ}\text{C}$ (Revised Manuscript, Page 14 and 15, Line 295-312). This observation indicates that hydrogenated silicon nanodomains and subsequent dehydrogenation of such nanodomains leading to the formation of silicon nanocrystals during annealing⁴, did not occur in the laser-exposed HSQ (i.e., the as-printed glass), indicating a low hydrogen content in those regions. Finally, we observed that the residual Si-H bonds in the as-printed glass were completely removed after annealing at only $150\text{ }^{\circ}\text{C}$, as seen in the corresponding Raman spectrum.

Inspired by this comment, we performed new nanoindentation experiments and analyses based on the Oliver-Pharr method¹ to obtain hardness and reduced elastic modulus for three different sample types. The first sample type consists of a microplate with a footprint of 20 μm x 20 μm and a thickness of about 2 μm , 3D-printed on a fused-silica substrate. The fused-silica substrates that we used here, and throughout the work, are JGS2 optical-grade fused quartz purchased from MicroChemicals. The second sample type consists only of a flat JGS2 fused-silica substrate for reference. Finally, to consider potential effects of the geometry of the plates on the indentation measurement results, we fabricated a third sample type by engraving microplates of the same dimensions as the 3D-printed plates into the same JGS2 fused-silica substrate using laser ablation. For the indentation measurements, we used two specimens of each sample type, one of the specimens was annealed at 900 $^{\circ}\text{C}$ for one hour, and the other one was not.

We have summarized the results of the nanoindentation experiments in Supplementary Table 3, and one representative loading-unloading curve for each sample type is shown in Supplementary Fig. 7. The results and the curves are also included below. The measured hardness and reduced elastic modulus of the flat JGS2 fused silica substrates before and after annealing at 900 $^{\circ}\text{C}$ were similar for both specimen which are about 9.5 GPa and 70 GPa, respectively. These values are in line with the expected values of fused silica². However, the measured hardness and reduced elastic modulus of laser-ablated fused-silica microplates before and after annealing were about 7.8 GPa and 65 GPa which are slightly lower than those of flat substrates. This shows that the geometry and dimensions of the microplates must be considered when interpreting the nanoindentation measurement results. Therefore, we will primarily compare the results of the 3D-printed microplates to those of laser-ablated fused-silica microplates in the following discussion since the laser-ablated samples are the most relevant reference samples because they are similar in geometry and dimensions to our 3D printed glass plates.

The measured hardness and reduced elastic modulus of the 3D-printed microplates were 2.4 ± 0.2 GPa and 40 ± 2 GPa, respectively, as printed. After annealing at 900 $^{\circ}\text{C}$ for 1 h, the hardness and reduced elastic modulus values reached 7.7 ± 0.6 GPa and 75 ± 2 GPa, respectively, i.e., similar values as the fused silica samples of the same geometry. The somewhat lower values of our as-printed glass are expected since we have observed residual hydrogen, hydroxyl groups,

and water content in its Raman spectrum. Also, we observed that the as-printed glass shrinks by a small amount (6.1 %) upon annealing. Interestingly, the hardness and reduced elastic modulus of our 3D-printed glass reach the values of the reference fused-silica microplates after annealing at 900 °C. This suggests that the residual components in our 3D-printed glass were completely removed, and that the glass was fully densified at an annealing temperature of 900 °C. This agrees with our observation that the Raman spectrum of the 3D-printed glass after annealing is indistinguishable from that of fused silica. This is also supported by our observation that the material shrinkage reached the ultimate value of 6.1% after annealing at 900 °C. Along with our EDS data and the fact that no pores were observed by TEM in the as-printed glass, down to the scale of a few nanometers, we conclude that the as-printed glass is low-density silica glass with a residual amount of hydrogen, hydroxyl groups, and water, and that annealing at 900 °C transforms it into fully densified silica glass that is indistinguishable from fused silica.

In the revised manuscript and supplementary information, we have included the new experimental results, analysis and discussion that we performed in response to this comment of the reviewer. Specifically, we have included hardness and reduced elastic modulus measured from the 3D-printed glass and references. We have also analyzed and discussed these results in depth in the related text passages, which now read:

Revised Manuscript, Page 4 and 5, Line 78-85:

“We show by Raman, energy-dispersive, and photoluminescence spectroscopy that the as-printed material is silica glass but, compared to fused silica glass, features a higher ratio of 4-membered silicon-oxygen rings in the network resulting from sub-picosecond laser exposure, photoluminescence, residual hydrogenated and hydroxyl species, and trace amounts of organic residuals. These features and residuals can be removed by a 900 °C annealing step, resulting in a low shrinkage of 6.1% of the 3D-printed structures and an increase of the hardness and reduced elastic modulus of the 3D-printed silica glass to values expected for fused silica glass.”

Revised Manuscript, Page 12, Line 230-249:

“The mechanical properties of the 3D printed silica glass are important for its use in many applications, including nanoelectromechanical systems (NEMS). With this in mind, we characterized the hardness and reduced elastic modulus of the 3D-printed silica glass by

performing nanoindentation measurements on 3D-printed microplates with a footprint of 20 μm by 20 μm and a thickness of about 2 μm . The measured hardness and reduced elastic modulus of the as-printed silica glass were 2.4 ± 0.2 GPa and 40 ± 2 GPa, respectively. After annealing at 900 $^{\circ}\text{C}$ for 1 h, the hardness and reduced elastic modulus increased to 7.7 ± 0.6 GPa and 75 ± 2 GPa, respectively, which are almost identical to the values we measured of reference samples consisting of fused-silica microplates of the same geometry (Supplementary Table 3 and Supplementary Fig. 7). The differences in the hardness and reduced elastic modulus between the as-printed glass and the fused silica reference sample are expected since we have observed residual hydrogen, hydroxyl groups, and water content in the Raman spectrum of the as-printed glass, and since the as-printed glass shrinks by up to (6.1 ± 0.8) % upon annealing (Fig. 2a, b). Our observation that the hardness and reduced elastic modulus of the 3D-printed glass after annealing at 900 $^{\circ}\text{C}$ reached the values of the fused-silica microplate reference samples, suggests that the residual components in our 3D-printed glass were completely removed, and that the glass was fully densified at an annealing temperature of 900 $^{\circ}\text{C}$. This is also consistent with our observations that the Raman spectrum of the 3D-printed glass after annealing is indistinguishable from that of fused silica, and that the material shrinkage reached its ultimate value of 6.1% after annealing at 900 $^{\circ}\text{C}$.”

Revised Manuscript, Page 23, Line 480-496:

“**Nanoindentation characterization.** Three types of samples were characterized. The first sample type consists of a microplate with a footprint of 20 μm x 20 μm and a thickness of about 2 μm , 3D-printed on a fused-silica substrate. The second sample type consists only of a flat fused-silica substrate for reference. Finally, to consider potential effects of the geometry of the plates on the indentation measurement results, we fabricated a third sample type by engraving microplates of the same dimensions as the 3D-printed plates into a fused-silica substrate using laser ablation. In all experiments, the fused-silica substrates were JGS2 optical-grade fused quartz purchased from MicroChemicals GmbH. For the indentation measurements, we used two specimens of each of the three sample types, one of the specimens was annealed at 900 $^{\circ}\text{C}$ for one hour, and the other specimen was not annealed. A nanoindenter (Hysitron TI 950 Triboindenter, Bruker) was used to measure the hardness and reduced elastic modulus of the samples via the Oliver–Pharr method¹. The indenter was equipped with a Berkovich tip and was calibrated using a

standard quartz reference sample with a hardness and elastic modulus of 9.3 and 69.6 GPa, respectively. All measurements were run in the load-controlled mode with set maximum loads that yielded indents that are sufficiently deep to avoid tip radius influence and do not exceed 200 nm, that is, not more than 10% of the thickness of the microplates, to avoid contributions from the substrate. On each sample, three measurements were performed.”

Revised Supplementary Information, Page 4:

Supplementary Table 3. Summary of nanoindentation characterization results.

Measurement results of our 3D-printed microplates, along with reference samples consisting of laser-ablated fused-silica microplates and flat fused-silica substrates. All sample types were measured without, and with annealing at 900 °C. For each sample, three measurements were performed. The measured hardness and reduced elastic modulus values are in the format of average \pm standard deviation of the three measurements of each sample.

Sample	Hardness (GPa)	Reduced elastic modulus (GPa)
3D-printed microplate without annealing	2.4 \pm 0.2	40 \pm 2
3D-printed microplate annealed at 900 °C	7.7 \pm 0.6	75 \pm 2
Laser-ablated fused-silica microplate	7.4 \pm 0.1	63.8 \pm 0.6
Laser-ablated fused-silica microplate annealed at 900 °C	8.1 \pm 0.5	66 \pm 1
Flat fused-silica substrate	9.44 \pm 0.05	69.7 \pm 0.7
Flat fused-silica substrate annealed at 900 °C	9.62 \pm 0.03	71.0 \pm 0.1

Revised Supplementary Information, Page 11:

Supplementary Fig. 7. Nanoindentation characterization of the 3D-printed glass and the fused silica glass reference samples. Loading-unloading curves of the samples listed in Supplementary Table 3. One of the measured three curve from each sample is plotted to demonstrate the behavior of the samples in the indentation experiments.

Reviewer's comment:

3. On a minor note:

1) in the introduction, the authors forgot to mention the possibility of fabricating silica glass structures using a hybrid approach (UV-assisted DIW), employing colloidal silica and TEOS as silica precursor (see: <https://doi.org/10.1016/j.addma.2022.102727>). This work should also be quoted and commented in the introduction;

Our response:

We thank the reviewer for bringing this recent and relevant work to our attention.

In the revised manuscript, we have included the work pointed out by the reviewer in the reference list and included a related discussion:

Revised Manuscript, Page 3, Line 40-46:

“To address this, additive manufacturing of silica glass by stereolithography^{5,6}, direct ink writing⁷⁻⁹, digital light processing¹⁰, and multiphoton polymerization¹¹⁻¹³ has been explored. Moreover, hybrid approaches that combine multiple manufacturing techniques and silica sources have been recently investigated¹⁴. Although 3D structures made of high-quality silica glass have been demonstrated, these approaches can, at best, resolve feature sizes of several tens of micrometers¹⁵, except for a recent study that has reported sub-micrometer resolution¹³.”

Reviewer's comment:

4. 2) in the introduction, the authors state: “This is because any substrate material involved in the printing process must withstand the thermal treatment, which essentially eliminates most materials of interest”. However, there is not necessarily a substrate material when using DLP, SLA and DIW (it depends on the form of the object how you attach it to the platform, for DLP and SLA, and the reported work by Dilla-Spears et al. shows the fabrication of self-standing lenses, while the work by De Marzi et al. reports the manufacturing of bulk pieces with an architected morphology). This sentence should therefore be corrected;

Our response:

We thank the reviewer for this comment and the opportunity to clarify this point. We agree that this sentence was not sufficiently clear and should be clarified. The message we were trying to convey with this sentence is that our approach by relieving the need for thermal treatments allows the direct integration of the 3D-printed silica-glass structures onto desired substrates or pre-manufactured devices that are sensitive to high-temperature treatments. In contrast, for other high-resolution 3D glass printing methods which all required debinding and sintering of the as-printed composites, it would only be possible to realize such integration of the 3D-printed structures by assembly in the end. The assembly would be exceedingly challenging especially in the cases of structures with sub-micrometer features, which in turn limits the integration capability of those approaches.

In the revised manuscript we have revised and clarified the related text passage which now reads:

Revised Manuscript, Page 3 and 4, Line 53-60:

“The mandatory sintering process at such elevated temperatures severely limits the application space and integration compatibility of these methods. This is because any substrate materials or pre-manufactured structures onto which the 3D-printed silica-glass structures are to be directly integrated must withstand the thermal treatment, which essentially eliminates most materials of interest. In other cases, final assembling of the 3D-printed structures and other substrates or structures required for the application would be necessary, which can be exceedingly challenging for structures at the scale of micrometers.”

Reviewer's comment:

- 5. 3) I do not see reported anywhere the source for the HSQ material; this should be clearly indicated in the manuscript.***

Our response:

We thank the reviewer for pointing this out. We agree that the way we reported the source of the HSQ was not sufficiently clear.

In the revised manuscript we have modified and clarified the related text passage which now reads:

Revised Manuscript, Page 19, Line 383-384:

“Next, HSQ solution (FOX16, Dow Corning, USA) which contains methyl isobutyl ketone (MIBK) and toluene as the solvents was drop-casted on the substrate.”

Reviewer's comment:

6. *The manuscript can be published after minor revisions.*

Our response:

We thank the reviewer for the thorough review of our work and the important comments that helped us to further improve our manuscript.

References

1. Oliver, W. C. & Pharr, G. M. An improved technique for determining hardness and elastic modulus using load and displacement sensing indentation experiments. *J. Mater. Res.* **7**, 1564–1583 (1992).
2. Gao, C. & Liu, M. Instrumented indentation of fused silica by Berkovich indenter. *J. Non Cryst. Solids* **475**, 151–160 (2017).
3. Olynick, D. L., Cord, B., Schipotinin, A., Ogletree, D. F. & Schuck, P. J. Electron-beam exposure mechanisms in hydrogen silsesquioxane investigated by vibrational spectroscopy and in situ electron-beam-induced desorption. *Journal of Vacuum Science & Technology B* **28**, 581 (2010).
4. Hessel, C. M., Henderson, E. J. & Veinot, J. G. C. An investigation of the formation and growth of oxide-embedded silicon nanocrystals in hydrogen silsesquioxane-derived nanocomposites. *Journal of Physical Chemistry C* **111**, 6956–6961 (2007).
5. Kotz, F. *et al.* Three-dimensional printing of transparent fused silica glass. *Nature* **544**, 337–339 (2017).
6. Toombs, J. T. *et al.* Volumetric additive manufacturing of silica glass with microscale computed axial lithography. *Science (1979)* **376**, 308–312 (2022).
7. Nguyen, D. T. *et al.* 3D-Printed Transparent Glass. *Advanced Materials* **29**, 1701181 (2017).
8. Destino, J. F. *et al.* 3D Printed Optical Quality Silica and Silica–Titania Glasses from Sol–Gel Feedstocks. *Adv. Mater. Technol.* **3**, 1700323 (2018).
9. Dylla-Spears, R. *et al.* 3D printed gradient index glass optics. *Sci. Adv.* **6**, 7429–7447 (2020).
10. Moore, D. G., Barbera, L., Masania, K. & Studart, A. R. Three-dimensional printing of multicomponent glasses using phase-separating resins. *Nature Materials* **2019** *19*:2 **19**, 212–217 (2019).
11. Kotz, F. *et al.* Two-Photon Polymerization of Nanocomposites for the Fabrication of Transparent Fused Silica Glass Microstructures. *Advanced Materials* **33**, 2006341 (2021).
12. Doualle, T., André, J.-C. & Gallais, L. 3D printing of silica glass through a multiphoton polymerization process. *Opt. Lett.* **46**, 364 (2021).
13. Wen, X. *et al.* 3D-printed silica with nanoscale resolution. *Nat. Mater.* **20**, 1506–1511 (2021).

14. de Marzi, A., Giometti, G., Erlen, J., Colombo, P. & Franchin, G. Hybrid additive manufacturing for the fabrication of freeform transparent silica glass components. *Addit. Manuf.* **54**, 102727 (2022).
15. Kotz, F., Risch, P., Helmer, D. & Rapp, B. E. High-Performance Materials for 3D Printing in Chemical Synthesis Applications. *Advanced Materials* **31**, 1805982 (2019).

Reviewer comments, second round

Reviewer #2 (Remarks to the Author):

In this well revised manuscript and Supplementary Information Authors have thoroughly addressed all my comments, therefore I recommend the publication in their present form.

Reviewer #3 (Remarks to the Author):

The authors revised their manuscript appropriately and replied to all the queries from the reviewers. The additional nano-indentation experiments confirmed that, indeed, a material akin to fused silica glass is obtained only after heating at 900°C. In my opinion, this detracts a bit from the original claim of the authors, that is that their material is a glass just after printing. However, the authors clearly explained the situation in the revised text, so the readers will be able to understand that only some of the characteristics of the material are similar to silica when the material is still in the 3D printed state, while other characteristics (i.e., mechanical properties and, I suppose, density and therefore refractive index) require a further annealing at high temperature. In any case, this is a very good work, well worth publishing. The manuscript can now be published as is. or revisions.

Point-by-Point Response to the Comments of the Reviewers

Reviewer #2:

Reviewer's comment:

- 1. In this well revised manuscript and Supplementary Information Authors have thoroughly addressed all my comments, therefore I recommend the publication in their present form.*

Our response:

We sincerely thank the reviewer for carefully reviewing our work and the constructive and insightful comments, which have helped us to improve the manuscript and supplementary information substantially.

Reviewer #3:

Reviewer's comment:

- 1. The authors revised their manuscript appropriately and replied to all the queries from the reviewers. The additional nano-indentation experiments confirmed that, indeed, a material akin to fused silica glass is obtained only after heating at 900°C. In my opinion, this detracts a bit from the original claim of the authors, that is that their material is a glass just after printing. However, the authors clearly explained the situation in the revised text, so the readers will be able to understand that only some of the characteristics of the material are similar to silica when the material is still in the 3D printed state, while other characteristics (i.e., mechanical properties and, I suppose, density and therefore refractive index) require a further annealing at high temperature. In any case, this is a very good work, well worth publishing. The manuscript can now be published as is.*

Our response:

We sincerely thank the reviewer for carefully reviewing our work and the constructive and insightful comments, which have helped us to improve the manuscript and supplementary information substantially.